# Resistance to cancer immunotherapy mediated by apoptosis of tumor-infiltrating lymphocytes

Jingjing Zhu[1,2,3], Céline G. Powis de Tenbossche[1,2], Stefania Cané[1,2,3], Didier Colau[1,2], Nicolas van Baren[1,2], Christophe Lurquin[1,2], Anne-Marie Schmitt-Verhulst [4], Peter Liljeström[5], Catherine Uyttenhove[1,2] & Benoit J. Van den Eynde [1,2,3]

Despite impressive clinical success, cancer immunotherapy based on immune checkpoint blockade remains ineffective in many patients due to tumoral resistance. Here we use the autochthonous TiRP melanoma model, which recapitulates the tumoral resistance signature observed in human melanomas. TiRP tumors resist immunotherapy based on checkpoint blockade, cancer vaccines or adoptive T-cell therapy. TiRP tumors recruit and activate tumor-specific CD8[+] T cells, but these cells then undergo apoptosis. This does not occur with isogenic transplanted tumors, which are rejected after adoptive T-cell therapy. Apoptosis of tumor-infiltrating lymphocytes can be prevented by interrupting the Fas/Fas-ligand axis, and is triggered by polymorphonuclear-myeloid-derived suppressor cells, which express high levels of Fas-ligand and are enriched in TiRP tumors. Blocking Fas-ligand increases the anti-tumor efficacy of adoptive T-cell therapy in TiRP tumors, and increases the efficacy of checkpoint blockade in transplanted tumors. Therefore, tumor-infiltrating lymphocytes apoptosis is a relevant mechanism of immunotherapy resistance, which could be blocked by interfering with the Fas/Fas-ligand pathway.

[1] Ludwig Institute for Cancer Research, Brussels B-1200, Belgium. [2] de Duve Institute, Université Catholique de Louvain, Brussels B-1200, Belgium. [3] Walloon Excellence in Life Sciences and Biotechnology, Brussels B-1200, Belgium. [4] Centre d'Immunologie de Marseille-Luminy, Aix Marseille Université, Inserm, CNRS, Marseille, France. [5] Department of Microbiology, Tumor, and Cell Biology, Karolinska Institutet, Stockholm SE-17177, Sweden. Jingjing Zhu, Céline G. Powis de Tenbossche and Stefania Cané contributed equally to this work. Correspondence and requests for materials should be addressed to B.J.V.d.E. (email: benoit.vandeneynde@bru.licr.org)

nhibitory antibodies against immune checkpoint molecules CTLA4 and PD1 induce durable tumor responses in a number of cancer patients, and have become standard of care in a number of metastatic malignancies. Yet clinical benefit of anti-CTLA4 and anti-PD1 remains limited to a fraction of patients and the priority is to understand why the majority of patients fail to respond. There is mounting evidence indicating that immunotherapy resistance is largely dependent on the tumor microenvironment, whose immunosuppressive nature is progressively shaped during the long-term process of tumor development in an immunocompetent host. There are numerous cellular and molecular mechanisms at play, and the challenge is to define those that are clinically relevant. This is traditionally based on preclinical studies, which often rely on murine transplanted tumor models. However, transplanted tumors do not recapitulate the tumor microenvironment as it progressively develops during the growth of an autochthonous tumor. This is better modeled using genetically engineered mouse models (GEMM), in which autochthonous tumors develop following the induction of oncogenic events within host tissues. However, most available GEMM models either do not express defined tumor antigens, precluding in-depth analysis of the anti-tumor immune responses in the course of immunotherapy, or express model antigens such as ovalbumin or viral antigens, which are highly immunogenic and do not reflect the poor immunogenicity of tumor antigens that are naturally expressed on human tumors. To circumvent these issues, we created the GEMM model of inducible melanomas expressing P1A, a defined mouse tumor antigen of the MAGE type[55], which we chose as the best representative of the clinically relevant group of human MAGE-type tumor antigens encoded by cancer-germline genes[1]. This model, named TiRP, is based on the tamoxifen-induced and Cre-lox-mediated induction of $H$-$Ras^{G12V}$ and deletion of $Ink4a/Arf$ in melanocytes[2]. The $H$-$Ras^{G12V}$ transgene is followed by an internal ribosome entry site (IRES) and the P1A coding sequence ($Trap1a$), so that all melanomas express P1A and are recognized by P1A-specific CD8[+] T cells directed against the dominant P1A epitope, a nine-amino acid peptide presented by H-2L[d2]. To ensure H-2L[d] presentation, we backcrossed the model from B6 to B10.D2 mice, which are B6 congenic for H-2[d]. This TiRP model now gives a high incidence of melanomas (±95%) with a short latency (30–60 days depending on the tamoxifen batch). Induced TiRP melanomas are initially well differentiated and highly pigmented (Mela tumors), but then rapidly undergo dedifferentiation resulting in white aggressive tumors (Amela), characterized by Epithelial-to-Mesenchymal Transition (EMT)-like and TGFβ signatures[3]. This mesenchymal transition is accompanied by abnormal inflammation, involving recruitment of immature myeloid cells, which appear to impair anti-tumor immunity[4]. The Amela tumor signature shares many genes with the innate anti-PD1 resistance (IPRES) signature that was recently described in human melanomas resisting anti-PD1 therapy, including EMT-like genes ($Axl$, $Twist2$, $Loxl2$), angiogenesis genes ($Vegfa$) and monocyte/macrophage recruitment genes ($Ccl2$, $Ccl7$)[3, 5]. The TiRP model therefore provides an ideal platform to study immunotherapy resistance in a manner that is more relevant than transplanted tumors and recapitulates human tumors resisting immunotherapy.

In this report, we confirm that TiRP tumors are resistant to different forms of immunotherapy, and we link this resistance to the ability of TiRP tumors to induce tumor-infiltrating lymphocytes (TIL) apoptosis. Anti-tumor T cells recruited to TiRP tumors quickly undergo apoptosis and disappear. Therefore, they fail to reject the tumor and to persist. This happens with autochthonous TiRP tumors, but not with isogenic transplanted tumors. TIL apoptosis is triggered by Fas-ligand, which is expressed by polymorphonuclear myeloid-derived suppressor cells (PMN-MDSC). The latter are enriched in induced TiRP tumors as compared to transplanted isogenic tumors. Administration of soluble Fas-Fc can prevent TIL apoptosis and improve the efficacy of immunotherapy.

## Results

**Induced TiRP melanomas resist immunotherapy.** Immunotherapy based on checkpoint inhibitors anti-CTLA4 and anti-PD1 proved unable to alter the growth of tamoxifen-induced TiRP tumors, whether given alone or in combination (Fig. 1a). Because TiRP tumors express MAGE-type tumor antigen P1A, we also tested a preventive vaccination against P1A using an established prime/boost immunization scheme based on P1A-recombinant adenovirus and Semliki Forest virus (SFV), which induces high levels of P1A-specific CD8[+] T cells and significant protection against a challenge injection of P1A-positive tumor cells[6]. However, immunization of TiRP mice before tumor induction with tamoxifen failed to prevent tumor onset, and this was not improved upon combination of vaccine with anti-CTLA4 and anti-PD1 (Fig. 1a). This was seemingly not due to loss of antigen expression, as the tumors that appeared in vaccinated mice still expressed high amounts of P1A transcripts (Supplementary Fig. 1). These results suggested a strong immunosuppressive mechanism preventing T-cell attack of TiRP tumors.

We then used H-2L[d]-P1A tetramers to monitor P1A-specific CD8[+] T cells in the spleen of vaccinated mice stimulated one week in vitro with the P1A peptide (LPYLGWLVF) (Fig. 1b)[49]. All vaccinated mice developed an anti-P1A CD8[+] response. In control mice without tumor, this response persisted until at least 70 days after vaccination. However, in mice developing a TiRP tumor, the P1A-specific CD8[+] T-cell response plummeted to the levels of unimmunized mice at day 70. Hence, the presence of a TiRP tumor induced complete disappearance of the P1A-specific anti-tumor CD8[+] response. We also observed that non-vaccinated mice developing a TiRP tumor spontaneously mounted a P1A-specific immune response, confirming that TiRP tumors express the P1A antigen and indicating that the tumor is immunogenic. However, as observed in vaccinated mice, this anti-tumor immune response did not persist. This lack of persistence was restricted to anti-tumor CD8[+] T cells, because when we immunized mice against both P1A and an irrelevant antigen named P91A (an H-2L[d]-restricted mutated peptide[7]), we observed that the CD8[+] response against the latter antigen persisted in tumor-bearing mice (Fig. 1c). Moreover, tumor-bearing mice were able to mount a normal primary CTL response against minor histocompatibility antigens (Supplementary Fig. 2), indicating that tumor development did not induce a general CTL unresponsiveness, as reported previously in a spontaneous sporadic tumor mouse model[8, 9]. In the latter model, increased levels of TGFβ1 were measured in the serum of tumor-bearing mice, and correlated with the general T-cell unresponsiveness observed after a long latency period[8, 9]. Therefore, we also measured TGFβ1 levels in the serum of tumor-bearing TiRP mice (Supplementary Fig. 3). Total TGFβ1 levels were high in both groups and increased in tumor-bearing mice as compared to tumor-free mice. However, the levels of active TGFβ1 were low and similar in both groups, indicating that most of the seric TGFβ1 is in its latent, inactive form. Altogether, our data do not suggest a lack of general T-cell responsiveness, but rather indicate a selective suppression of tumor-specific CTL in tumor-bearing TiRP mice.

**Induced but not transplanted tumors resist adoptive cell therapy.** To better track the fate of tumor-specific CD8[+] T cells in this model, and to evaluate another clinically relevant immunotherapy approach, we resorted to adoptive transfer of P1A-

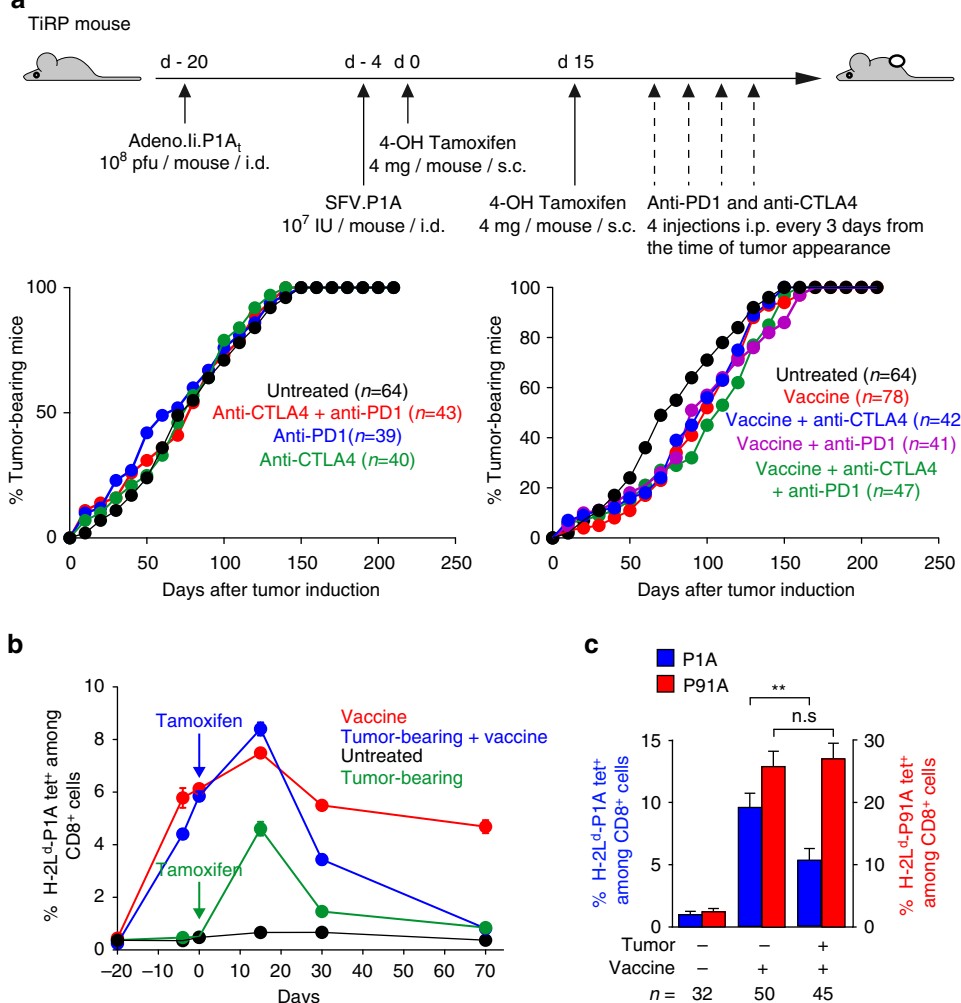

**Fig. 1** Induced TiRP tumors resist immunotherapy. **a** Schedule for tumor induction and immunotherapy. TiRP mice (B10.D2;Ink4a/Arf$^{flox/flox}$;TiRP$^{+/+}$) injected twice with 4 mg 4OH-tamoxifen on day 0 and 15, were also injected as indicated with anti-CTLA4 (4 × 40 μg) and/or anti-PD1 (4 × 200 μg) and/or a prime/boost vaccine regimen of recombinant adenovirus (Adeno.Ii.P1A$_t$) and Semliki Forest virus (SFV.P1A) encoding the MAGE-type tumor antigen P1A. Tumor appearance and mice survival were monitored. The figure represents the cumulative data of three experiments. **b** Tumor-bearing TiRP mice and control mice (B10.D2;Ink4a/Arf$^{flox/flox}$ mice, which have the same genetic background as TiRP mice but lack the TiRP transgene) were immunized with the P1A vaccine as above, and the P1A-specific CD8$^+$ T-cell response was monitored by FACS in spleen cells stimulated one week with P1A-peptide pulsed spleen cells, using antibodies to CD3ε and CD8α, and H-2L$^d$-P1A tetramers. Five mice were analyzed at each time point for each group. **c** TiRP mice were immunized simultaneously against P1A as above and against the irrelevant H-2L$^d$-restricted antigen P91A (mutated peptide) by intramuscular injections of P91A peptide in AS15 adjuvant[52] one week apart during 1 month. Mice were then treated with 4OH-tamoxifen. At the time of tumor appearance, spleen cells were stimulated one week with P1A or P91A peptide-pulsed spleen cells, and P1A- or P91A-specific CD8$^+$ T cells were quantified by FACS analysis using the relevant H-2L$^d$ tetramers. Results are expressed as mean + s.e.m. Unpaired t-test, two-tailed **c**, *$P < 0.05$, **$P < 0.01$

specific CD8$^+$ T cells isolated from mice transgenic for the anti-P1A T-cell receptor (TCR) (TCRP1A mice in the Rag1$^{-/-}$ B10.D2 background)[10]. Mice bearing induced Amela TiRP tumors received an intravenous injection of 10 million CD8$^+$ T cells isolated from TCRP1A mice and stimulated 4 days in vitro (Fig. 2a). Strikingly, this adoptive cell therapy (ACT) was unable to alter the growth of induced TiRP tumors (Fig. 2b). In parallel, we used tumor cell line T429, which was previously established from an induced Amela TiRP tumor and grown in vitro[4]. We established isogenic transplanted tumors by subcutaneous injection of cells from a T429 clone, and we treated these mice with ACT as above. In contrast to induced tumors, those transplanted tumors were efficiently rejected by adoptively transferred TCRP1A CD8$^+$ T cells (Fig. 2c). Similar results were obtained with distinct clones from line T429 and with tumors transplanted in the subcutaneous (Fig. 2d) or the intradermal space, which is

the natural niche for melanoma development (Fig. 2e). Because of the identical genetic background of the tumor cells, this striking difference between induced and transplanted tumors likely resulted from differences in the tumor microenvironment, and underlines the relevance of autochthonous tumor models as opposed to transplanted tumor models for preclinical evaluation of cancer immunotherapy.

We then analyzed the persistence of TCRP1A CD8$^+$ T cells in the spleen and lymph nodes of mice 22 days after ACT, using H-2L$^d$-P1A tetramers ex vivo. P1A-specific CTL were easily detected in tumor-free mice and in mice bearing transplanted tumors. However, they were undetectable in mice bearing an induced tumor (Fig. 2f). We then used TCR-specific clonotypic PCR as a more sensitive approach, but we also failed to detect TCRP1A T cells after 22 days in the spleen of mice bearing induced tumors, while we easily detected them in tumor-free and tumor-

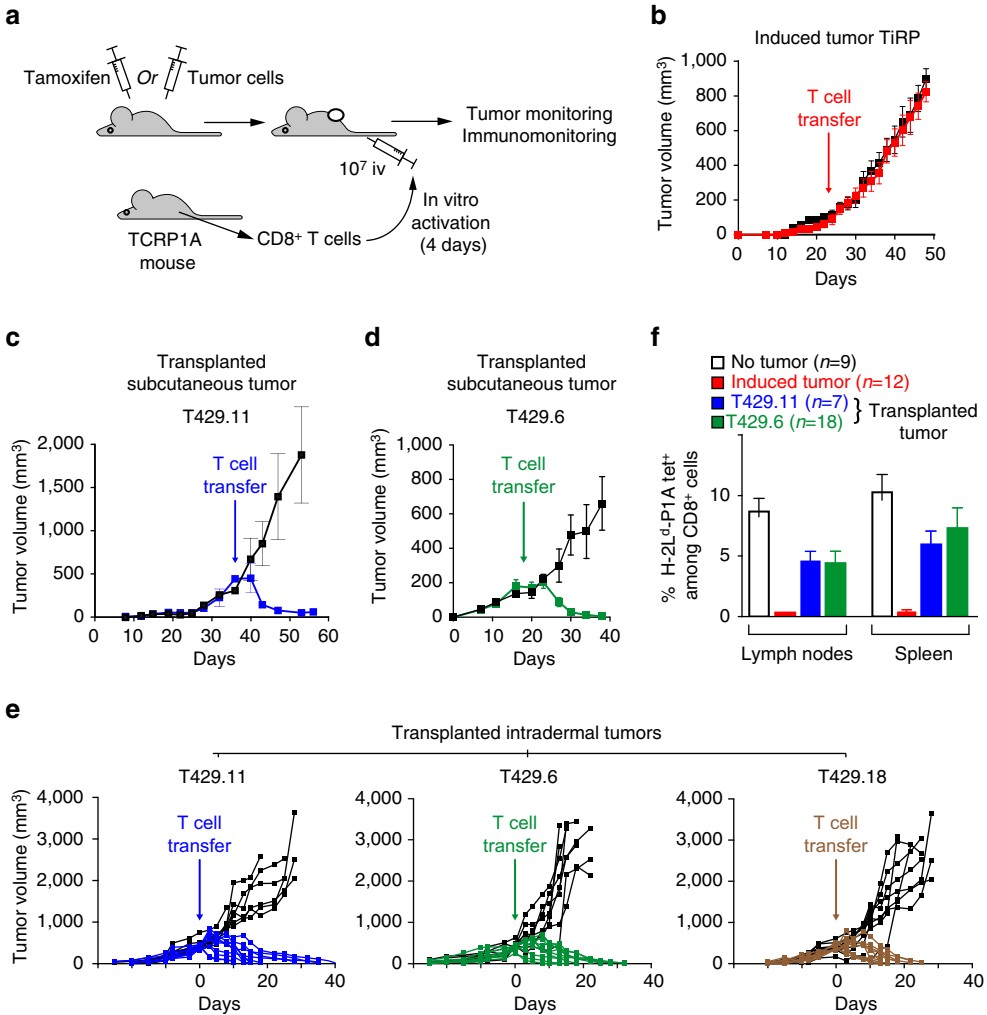

**Fig. 2** Rejection of transplanted but not induced autochthonous tumors after adoptive transfer of tumor-specific CD8[+] T cells. **a** Adoptive transfer protocol. Mice bearing induced autochthonous Amela TiRP tumors or transplanted isogenic tumors received intravenously 10[7] CD8[+] T cells that were isolated from TCRP1A B10.D2 mice and activated 4 days in vitro by co-incubation with lethally irradiated cells expressing P1A and B7-1 (L1210.P1A.B7-1). **b** TiRP mice bearing induced Amela tumors 22–24 days after 4OH-tamoxifen injection received adoptive transfer (n = 32) of 10[7] TCRP1A CD8[+] T cells activated in vitro for 4 days (*red symbols*). *Black symbols* show control mice (n = 30) receiving no T cells. Tumor volume is shown. (Data accumulated from two identical experiments). **c** Mice (B10.D2;Ink4a/Arf[flox/flox]) were injected subcutaneously with 2 × 10[6] cells from clone 11 of isogenic tumor line T429, which had previously been adapted to culture from an induced Amela TiRP tumor. After 36 days, mice received an intravenous injection of 10[7] activated TCRP1A CD8[+] T cells (*blue symbols*). Control mice received no T cells (*black symbols*). Tumor volume was monitored. Results are shown from one representative experiment (n = 6/group) out of at least three performed. **d** Same as in panel **c** but using another clone (T429.6) from tumor line T429. Mice received 10[7] activated TCRP1A CD8[+] T cells (*green symbols*) or not (*black symbols*) 18 days after tumor injection. Results are shown from one representative experiment (n = 6–7/group) out of at least three performed. **e** B10.D2;Ink4a/Arf[flox/flox] mice were injected intradermally (2 × 10[6] cells) with three distinct clones of tumor line T429. When tumors reached a size of about 400 mm[3], mice received an intravenous injection of 10[7] activated TCRP1A CD8[+] T cells (*colored curves*). *Black symbols* indicate control mice that received no T cells. Tumor growth was monitored. Individual growth curves are shown (8–10 mice/group). **f** Mice treated as in *panels* **b**–**d** were killed 22 days after T-cell transfer, and cells from the spleen and the draining lymph nodes were tested ex vivo (i.e. without in vitro stimulation) for the presence of P1A-specific T cells by FACS using H-2L[d]-P1A tetramer. The number of mice analyzed is indicated for each group. Results are expressed as mean ± s.e.m

transplanted mice (Supplementary Fig. 4a). At this time point, TCRP1A T cells were also absent in most of the induced tumors themselves (Supplementary Fig. 4a). These results indicated that tumor-specific T cells were actively deleted in mice bearing induced Amela TiRP tumors, while they were not in mice bearing transplanted tumors.

**Induced tumors trigger apoptosis of tumor-specific T cells.** To understand the fate of tumor-specific T cells, we then analyzed mice earlier, 4 days after ACT. At this time, both induced and transplanted tumors were infiltrated by CD8[+] T cells (Fig. 3a),

and transferred TCRP1A CD8[+] T cells were easily detected in induced tumors by clonotypic PCR (Supplementary Fig. 4b). Ex vivo tetramer analysis further showed that transferred T cells were enriched in the tumors as compared to the spleen and draining lymph nodes, in both induced and transplanted tumors (Fig. 3b). These results indicated that transferred TCRP1A CD8[+] T cells effectively migrated into the induced tumors in the first days after adoptive transfer. Furthermore, those tumor-infiltrating T cells expressed activation marker CD69, indicating that they recognized the P1A tumor antigen on tumor cells (Fig. 3c). To further evaluate the functional capacity of

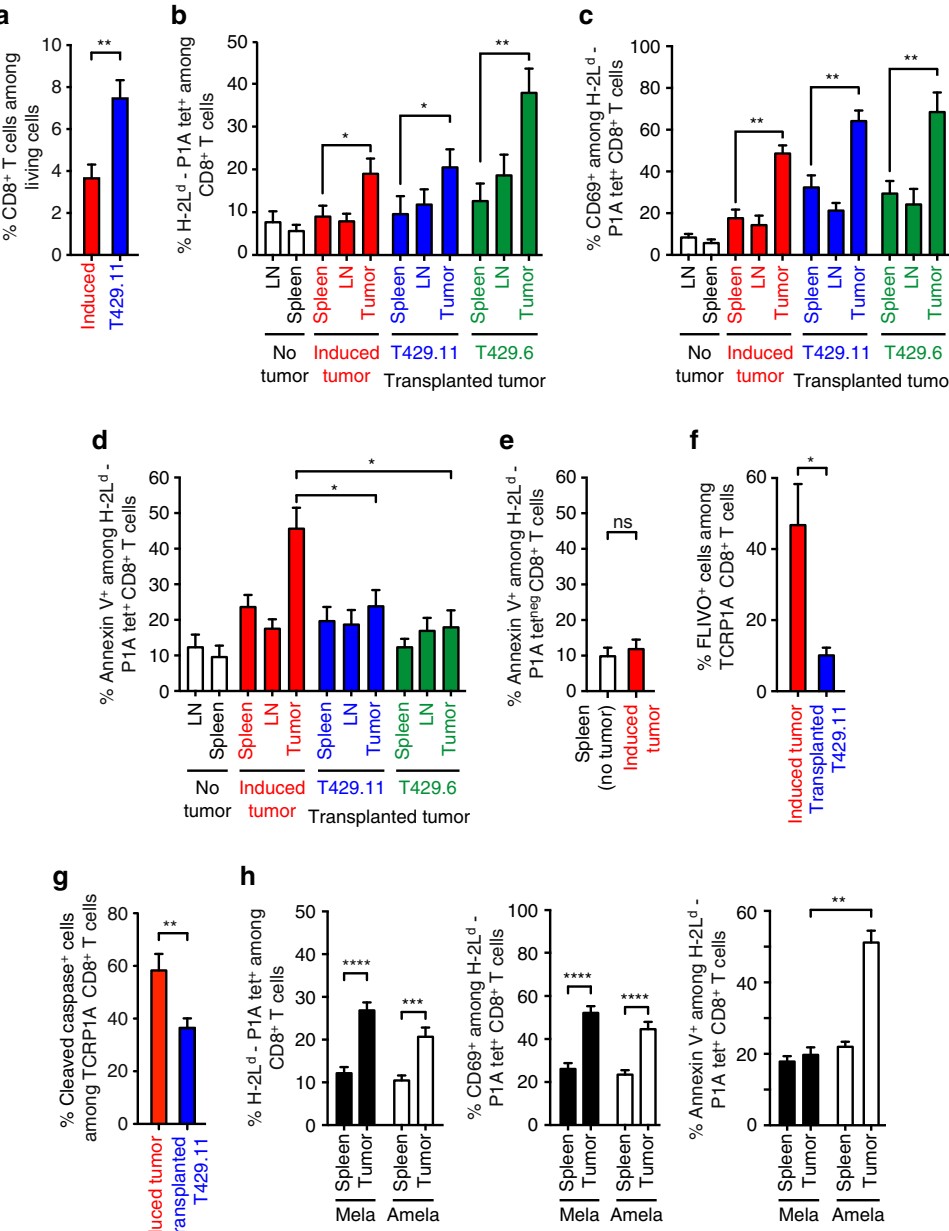

**Fig. 3** In vivo apoptosis of tumor-infiltrating CD8[+] T cells 4 days after transfer. **a** Mice bearing induced ($n = 14$) or transplanted ($n = 13$) tumors (500 mm³) received adoptive transfer of TCRP1A CD8[+] T cells as in Fig. 2. Four days later, tumors were analyzed by FACS ex vivo for CD8[+] T cells among living cells. **b–d** Draining lymph nodes (LN), spleens and tumors from tumor-bearing mice were analyzed 4 days after adoptive transfer of TCRP1A CD8[+] T cells by ex vivo FACS staining for CD8 and H-2L[d]-P1A tetramers **b**, CD69 **c**, and with Annexin V **d** (for **b–d**: $n = 75$ mice for induced tumors, $n = 75$ mice for T429.11, $n = 32$ mice for T429.6, $n = 12$ for tumor-free mice). **e** H-2L[d]-P1A tetramer-negative CD8[+] T cells infiltrating induced tumors or spleens from tumor-free mice were stained for Annexin V. Mice were identical to **b–d** (induced tumors: $n = 75$, spleens from tumor-free mice: $n = 12$). **f** Mice bearing induced ($n = 4$) or transplanted ($n = 3$) tumors (500 mm³) were transferred with activated TCRP1A CD8[+] T cells. Four days after transfer, they received an i.v. injection of FLIVO (inhibitor-based pan-caspase probe) 4 h before killing. Apoptosis of TCRP1A CD8[+] T cells was evaluated ex vivo by FACS staining for FLIVO. Mice receiving a non-targeting FLIVO control dye showed no staining of TCRP1A CD8[+] T cells. **g** Slices (300 μm) of fresh tumor tissues were incubated with CMAC-stained TCRP1A CD8[+] T cells for 24 h. Cryosections (7 μm) were stained for apoptosis using inhibitor-based active pan-caspase marker FLICA, and scanned with a MIRAX digital microscope. Data were quantified using Biopix software ($n = 5$ mice/group; three sections analyzed per mouse). **h** TiRP mice bearing either pigmented (Mela) or unpigmented (Amela) induced tumors were treated and analyzed as in **b–d** ($n = 20$ mice/group). Results are expressed as mean + s.e.m. Unpaired t-test, two-tailed **a–h**, *$P < 0.05$, **$P < 0.01$, ***$P < 0.001$. ****$P < 0.0001$

transferred T cells in the early days after ACT, we tested their ability to kill P1A-peptide pulsed target cells in an in vivo killing assay performed 3 days after transfer (Supplementary Fig. 5). We observed efficient killing of P1A-positive targets in mice bearing induced tumors, even though it was slightly reduced as compared with tumor-free mice. Thus, the results so far indicated that

transferred TCRP1A CD8[+] T cells retained their functional capacity, infiltrated the induced tumors, recognized their antigen and became activated. This raised the question: why did they fail to reject the induced tumors?

We then observed that 4 days after adoptive transfer, a high proportion of P1A-specific CD8[+] T cells infiltrating the induced

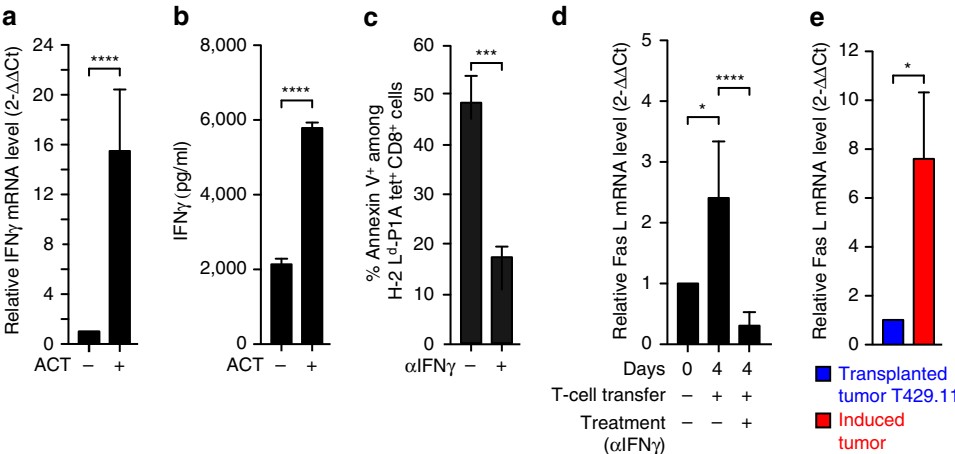

**Fig. 4** Role of IFNγ in triggering apoptosis of tumor-specific CD8+ T cells. **a** Quantitative RT-PCR analysis of IFNγ mRNA expression in induced TiRP tumor tissues collected 4 days after ACT. Results normalized to β-actin are expressed relative to the level measured in control tumors that did not receive ACT (controls: n = 20; ACT: n = 24). **b** Fresh homogenates from induced tumors collected 4 days after ACT were cultured in vitro for 24 h and supernatants were tested by ELISA for the presence of IFNγ (controls: n = 22; ACT: n = 34). **c** Tumor-bearing mice received 0.5 mg neutralizing anti-IFNγ antibody i.p. 1 day before transfer of activated TCRP1A CD8+ T cells. Four days after transfer, dissociated tumor tissues were analyzed ex vivo by FACS for apoptosis of TCRP1A CD8+ T cells (controls: n = 23; anti-IFNγ: n = 9). **d** Quantitative RT-PCR analysis of FasL mRNA expression in induced TiRP tumor tissues collected 4 days after transfer of activated TCRP1A CD8+ T cells preceded or not by injection of neutralizing anti-IFNγ antibody as in **a–c**. Results normalized to β-actin mRNA level are expressed relative to the level measured in control tumors that did not receive TCRP1A CD8+ T-cell transfer (controls: n = 12; T-cell transfer: n = 11; T-cell transfer + anti-IFNγ: n = 8). **e** Quantitative RT-PCR analysis of FasL mRNA expression in induced TiRP tumor tissues (n = 19) as compared with T429.11 transplanted tumor tissues (n = 14). Results normalized to Gapdh are expressed relative to the level measured in transplanted tumors. Results are expressed as mean ± s.e.m. Unpaired t-test, two-tailed, *P < 0.05, **P < 0.01, ***P < 0.001. ****P < 0.0001

tumors were apoptotic, as defined by staining with Annexin V (Fig. 3d). This was not the case in transplanted tumors (Fig. 3d). The non P1A-specific CD8+ T cells that infiltrated induced tumors were not stained with Annexin V (Fig. 3e). This suggested that induced tumors resisted immune rejection by triggering apoptosis of tumor-specific T cells. Because of the high background staining observed in spleen and lymph nodes with Annexin V, we used a series of other approaches to reveal apoptosis. 4 days after T-cell transfer, we injected mice with FLIVO, a fluorescent inhibitor-based pan-caspase marker that selectively binds active caspases and was developed for in vivo use[11]. After 4 h, we killed the mice and analyzed TIL by FACS after staining with H-2Ld-P1A tetramer. We observed that approximately 50% of P1A-specific TIL were stained with FLIVO in induced tumors, while only about 10% were in transplanted tumors (Fig. 3f). We also analyzed sections of those tumors by immunofluorescence, monitoring cells positive for FLIVO and for CellTracker Blue CMAC Dye, which was used to stain T cells before adoptive transfer. This confirmed the higher proportion of apoptotic TCRP1A CD8+ T cells in induced tumors (Supplementary Fig. 6a). Similar results were obtained with tumors from mice adoptively transferred but not treated with FLIVO, which were stained with antibodies to cleaved caspase-3 and to V-alpha8, the alpha chain of the TCR used in TCRP1A CD8+ T cells (Supplementary Fig. 6b, c). Lastly, we tried to recapitulate apoptosis induction by collecting tumors ex vivo and incubating freshly cut tumor slices with CMAC-labeled TCRP1A CD8+ T cells for 24 h. We then stained cryosections of these tumor slices with FLICA, a fluorescent inhibitor-based marker of activated pan-caspases, and again we observed more apoptotic TCRP1A CD8+ T cells in induced as compared with transplanted tumors (Fig. 3g).

We then performed ACT in pigmented (Mela) and unpigmented (Amela) induced TiRP tumors, and compared P1A-specific CD8+ TIL 4 days after transfer. We observed equal T-cell infiltration and activation, but T-cell apoptosis was restricted to Amela tumors, suggesting that the ability to induce T-cell apoptosis was linked to the inflammatory microenvironment that is typical of Amela tumors in this model (Fig. 3h)[4].

**TIL apoptosis is dependent on Fas-ligand.** We then tried to understand what triggered apoptosis of anti-tumor T cells in induced tumors. Interferon-gamma (IFNγ) can play a dual role in the course of the immune response, contributing to an efficient response by promoting effector T-cell differentiation and MHC class I expression on the one hand, but also, on the other hand, inducing inhibitory molecules that contribute to the negative feedback of the immune response, including IDO, FasL and PDL1[12–15]. Four days after T-cell transfer, induced tumors contained high levels of IFNγ mRNA and secreted protein, confirming the functionality of transferred T cells (Fig. 4a, b). To determine whether this IFNγ played a role in the induction of T-cell apoptosis, we injected an IFNγ-neutralizing antibody into mice bearing induced tumors 1 day before ACT. Four days after ACT, we observed a strong reduction in the number of apoptotic TILs, indicating the involvement of IFNγ in triggering T-cell apoptosis at the tumor site (Fig. 4c).

A number of genes induced by IFNγ encode proteins that can trigger lymphocyte apoptosis. These include IDO[12, 15], PDL1[12, 16] and Fas-ligand[17]. IDO was only weakly expressed in this model, and only barely increased after T-cell transfer. In contrast, PDL1 was well expressed in this model[4]. However, it was not involved in TIL apoptosis because the injection of an anti-PD1 blocking antibody failed to prevent apoptosis of transferred T cells (Supplementary Fig. 7a). This was in line with the lack of therapeutic effect of PD1 blockade in this model (Fig. 1a). Expression of Fas-ligand (FasL) transcripts was observed in TiRP tumors, increased after T-cell transfer, and was dramatically reduced after IFNγ neutralization (Fig. 4d). Moreover, FasL expression was higher in induced TiRP tumors as compared to transplanted T429 tumors (Fig. 4e). We therefore investigated the Fas-FasL pathway as potentially involved in triggering apoptosis of tumor-infiltrating T cells. We first used siRNA to silence Fas in TCRP1A CD8+ T cells and make them insensitive to FasL-

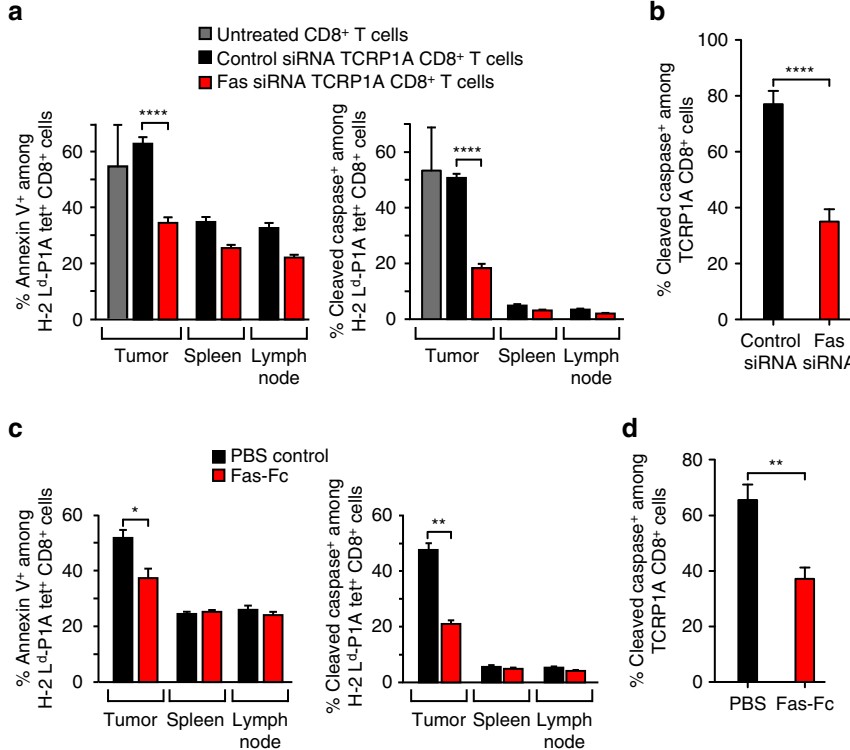

**Fig. 5** TIL apoptosis is triggered by FasL. **a** Tumor-bearing TiRP mice were transferred with $10^7$ activated TCRP1A CD8$^+$ T cells treated with either control siRNA or Fas siRNA. Four days later, tumor tissues, draining lymph nodes and spleen were analyzed ex vivo by FACS for apoptosis of TCRP1A CD8$^+$ T cells, using Annexin V (*left*) or pan-caspase marker FLICA (*right*) (n = 13 per group). **b** Fresh slices of induced TiRP tumors were incubated 24 h in vitro with TCRP1A CD8$^+$ T cells treated with control or Fas siRNA. Apoptosis of TCRP1A CD8$^+$ T cells was evaluated as in Fig. 3g (n = 5 mice/group; three sections analyzed per mouse). **c** Tumor-bearing TiRP mice transferred with $10^7$ activated TCRP1A CD8$^+$ T cells received daily injections of 150 μg soluble Fas-Fc starting 1 day before T-cell transfer. Four days later, tumor tissues, lymph nodes and spleen were analyzed ex vivo by FACS for apoptosis of P1A-specific CD8$^+$ T cells, using Annexin V (*left*) or active pan-caspase marker FLICA (*right*). (n = 13 per group). **d** Fresh slices of induced TiRP tumors pre-incubated 4 h with soluble Fas-Fc (10 μg/ml) were incubated in vitro with TCRP1A CD8$^+$ T cells and soluble Fas-Fc (10 μg/ml). Apoptosis was measured as in Fig. 3g (n = 5 mice/group; three sections analyzed per mouse). Unpaired *t*-test, two-tailed (**a-d**). Results are expressed as mean + s.e.m., *$P < 0.05$, **$P < 0.01$, ***$P < 0.001$. ****$P < 0.0001$ In figure 5 the panel (i) is explained in the legend but not mentioned in the figure. Please check. We changed (a-i) to (a-d)

mediated apoptosis, and we optimized the conditions so that Fas silencing was effective for at least 7 days (Supplementary Fig. 8). We then transferred Fas-silenced TCRP1A CD8$^+$ T cells into mice bearing induced TiRP tumors, and analyzed TIL apoptosis after 4 days, by FACS staining with Annexin V and for activated caspases (FLICA). We observed a strong reduction of apoptosis of Fas-silenced TIL (Fig. 5a). We obtained similar results when we incubated Fas-silenced TCRP1A CD8$^+$ T cells 24 h in vitro with fresh slices of induced tumors (Fig. 5b). To confirm this result, we then used soluble Fas-Fc to neutralize FasL in vivo. We injected mice bearing induced tumors with 150 μg Fas-Fc every day starting 1 day before ACT. Again we observed a clear reduction in the number of apoptotic TILs after 4 days (Fig. 5c). Fas-Fc also reduced apoptosis of TCRP1A CD8$^+$ T cells incubated in vitro with fresh slices of induced tumors (Fig. 5d). We conclude that the apoptosis of anti-tumor CD8$^+$ T cells observed in induced tumors is mostly driven by FasL. Tumor necrosis factor alpha (TNFα) is another factor known to induce T-cell apoptosis[18]. Even though TNFα is not clearly induced by IFNγ, we tested its involvement in T-cell apoptosis in the TiRP model by injecting an anti-TNFα neutralizing antibody 1 day before and 2 days after ACT. We observed no reduction of T-cell apoptosis on day 4 after ACT (Supplementary Fig. 7b). These results supported a dominant role of FasL in triggering T-cell apoptosis in the TiRP model.

**PMN-MDSC trigger TIL apoptosis**. We then compared the cellular composition of the tumor tissue of induced (Amela) and

transplanted tumors. Both tumor types contained CD45$^+$ cells (hematopoietic origin), which were more abundant in induced tumors. This likely resulted from a significant enrichment of induced tumors in polymorphonuclear-myeloid-derived suppressor cells (PMN-MDSC) ($P < 0.0001$, *t*-test), while monocytic myeloid-derived suppressor cells (M-MDSC) were equally represented in both tumor types (Fig. 6a). MDSC are a subset of immature myeloid cells that can be recruited at tumor sites and display immunosuppressive activity[19]. This PMN-MDSC enrichment was observed in Amela but not in Mela tumors (Fig. 6b). As compared with M-MDSC, PMN-MDSC produced more ROS, expressed less iNOS and produced less NO, in line with previous reports[20, 21] (Supplementary Fig. 9). They also expressed higher levels of arginase, another immunosuppressive factor that is often expressed in M-MDSC but also in PMN-MDSC[20–23]. We confirmed the immunosuppressive function of these PMN-MDSC by showing their ability to suppress in vitro proliferation and cytolytic activity of anti-tumor CD8$^+$ T cells at a 1/1 ratio (Fig. 6c, b). Moreover, when we established transplanted tumors by co-injecting T429.11 tumor cells with PMN-MDSC (isolated from induced tumors) at a 4 to 1 ratio and then treated mice with ACT, the co-injected tumors resisted immune rejection, while control tumors were rejected (Fig. 6e). These results confirmed the key role of PMN-MDSC in resistance of tumors to immunotherapy in this model. To explain the higher recruitment of PMN-MDSC in induced as compared with transplanted tumors, we measured the expression of a series of cytokines and

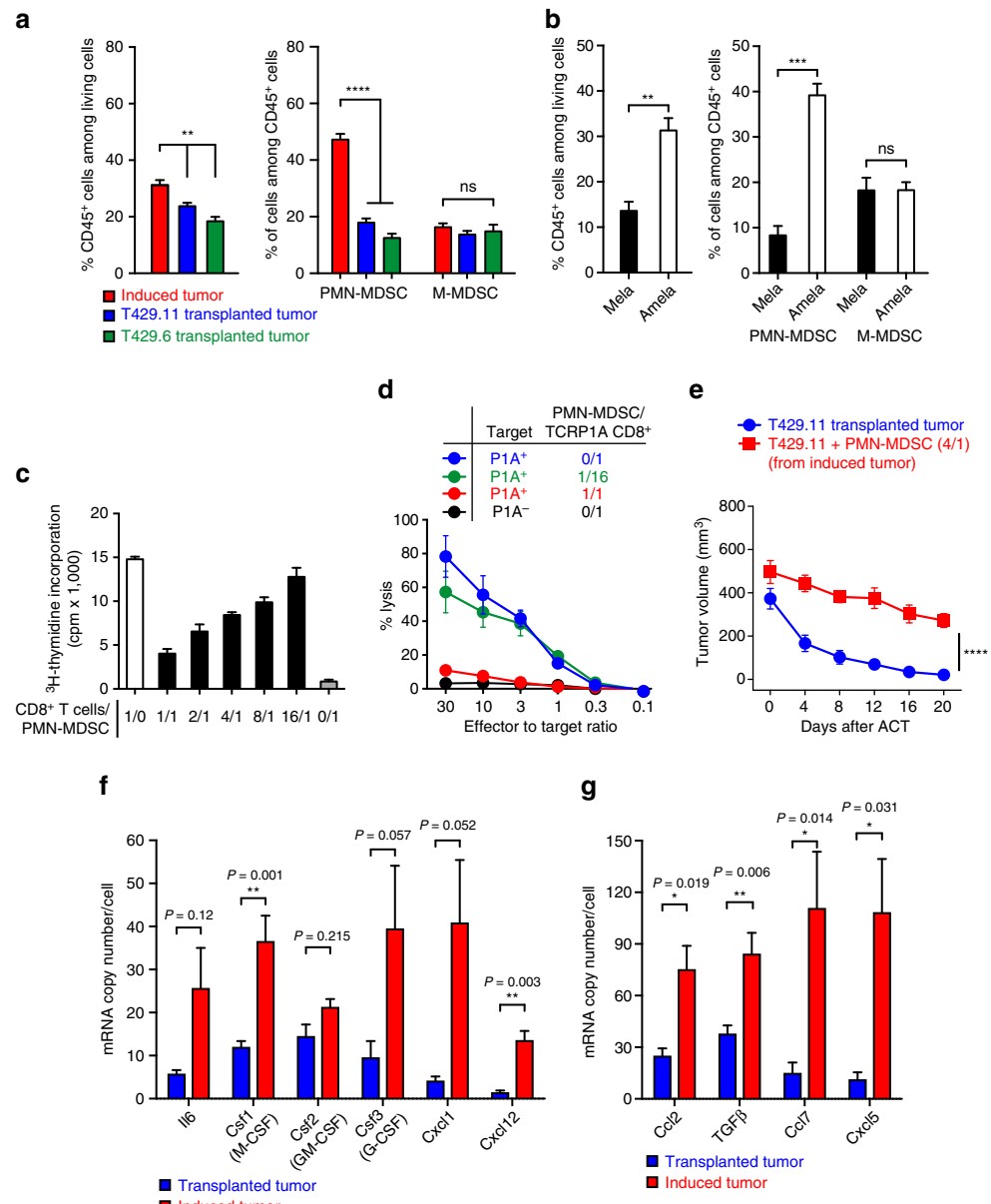

**Fig. 6** PMN-MDSC are enriched in induced Amela TiRP tumors. **a** Cellular analysis of the tumor tissue. Induced Amela tumors ($n = 27$) and transplanted T429.11 ($n = 25$) or T429.6 ($n = 18$) tumors (500 mm³) were homogenized and analyzed by FACS for the proportion of CD45⁺ cells (*left panel*) and MDSC of the polymorphonuclear type (PMN-MDSC: Gr-1ʰ, CD11b⁺, Ly6C⁻/ˡᵒ Ly6G⁺) or the monocytic type (M-MDSC: Gr-1ˡᵒ/ⁱⁿᵗ, CD11b⁺, Ly6Cʰ and Ly6G⁻). **b** Same analysis as in Fig. 5a comparing pigmented (Mela, $n = 16$) and unpigmented (Amela, $n = 16$) induced TiRP tumors. **c** PMN-MDSC isolated from induced Amela TiRP tumors were co-cultured with activated TCRP1A CD8⁺ T cells for 3 days. CD8⁺ T-cell proliferation was evaluated by measuring ³H-thymidine incorporation. Two independent experiments, in triplicates. **d** PMN-MDSCs isolated from induced Amela TiRP tumors were co-cultured for 3 days with activated TCRP1A CD8⁺ T cells. CD8⁺ T cells were then isolated and their ability to kill P1A-positive P815 cells (clone P511) was evaluated in a standard chromium release assay. Two independent experiments, in triplicates. **e** Melanoma cells T429.11 were mixed with PMN-MDSC isolated from induced Amela TiRP tumors at a 4/1 ratio and injected into the left flank of B10.D2;Ink4a/Arf^flox/flox mice. The right flank of the mice received the tumor cells without MDSC. ACT was performed when the left tumor size reached around 500 mm³ ($n = 8$). **f** and **g** Expression of cytokines and chemokines potentially involved in MDSC recruitment was analyzed by quantitative RT-PCR analysis in tumor tissues from induced Amela TiRP tumors ($n = 19$) and from transplanted T429.11 tumors ($n = 14$). *Gapdh* was used as an endogenous control to normalize each sample. Results are expressed as mean ± s.e.m. Unpaired *t*-test, two-tailed **a**, **b**, **f**, **g**. Two-way ANOVA **e**. *$P < 0.05$, **$P < 0.01$, ***$P < 0.001$. ****$P < 0.0001$

chemokines known to play a role in the recruitment or differentiation of MDSC[24, 25]. We found that most of them were expressed at a higher level in induced tumors as compared with transplanted tumors (Fig. 6f, g). Among these factors, Csf3, Cxcl1 and Cxcl5 are known to specifically recruit PMN-MDSC[24–26]. Although Ccl2, Cxcl12 and TGFβ are mostly known for their ability to recruit M-MDSC, they can also promote accumulation

of PMN-MDSC in some tumor settings[21, 24–26]. In addition, it has been suggested that M-MDSC recruited at the tumor site can be subsequently converted into PMN-MDSC in the tumor microenvironment[27]. These results could therefore explain the accumulation of PMN-MDSC in induced Amela TiRP tumors, and corroborate our previous observations of increased expression of such factors, including Ccl2, Cxcl5 and Ccl7, in Amela TiRP

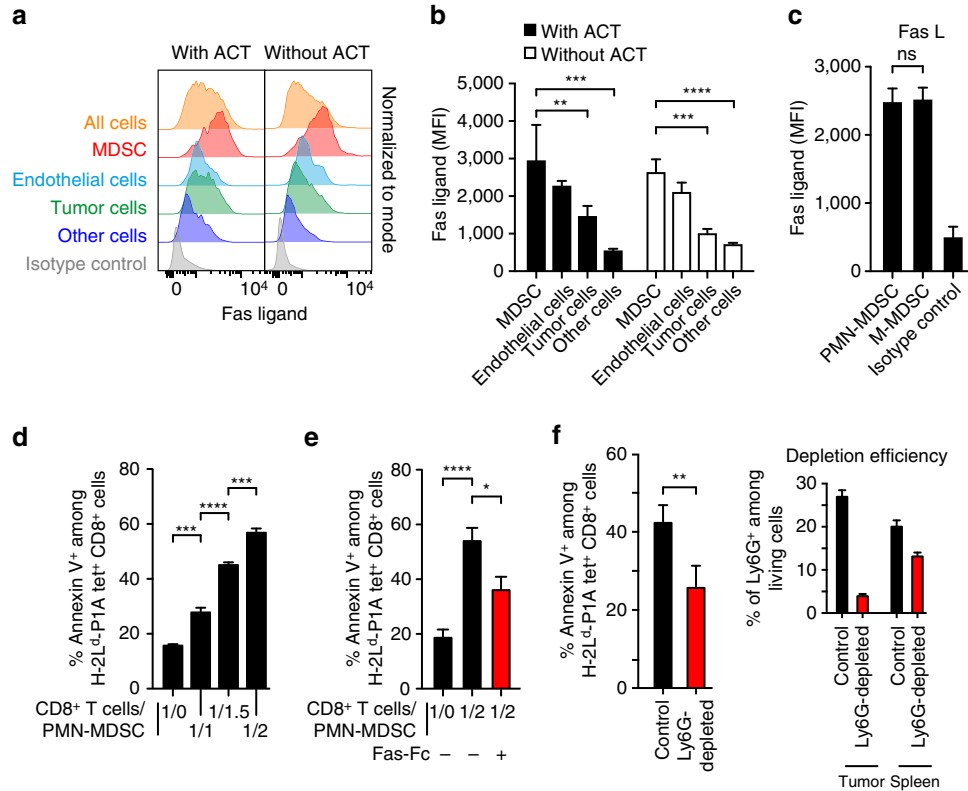

**Fig. 7** PMN-MDSC induce CD8[+] T-cell apoptosis through Fas-ligand. **a** FasL expression was analyzed by FACS on cell homogenates from two representative induced Amela TiRP tumors, having received ACT or not 4 days earlier. *Top panel*: whole cell population. *Second panel*: MDSC (Gr1[+] CD11b[+]). *Third panel*: Endothelial cells (CD31[+]). *Fourth panel*: Tumor cells (P1A[+], CD45[-]). *Fifth panel*: Other cells (Gr1[-], CD11b[-], CD31[-], P1A[-]). *Sixth panel*: isotype control. **b** Mean fluorescence intensity (MFI) of FasL expression of indicated cells obtained as in Fig. 6a (induced tumor with ACT: $n = 12$; induced tumor without ACT: $n = 12$). **c** Comparison of mean fluorescence intensity (MFI) of FasL expression by M-MDSC and PMN-MDSC (induced tumors without ACT: $n = 8$). **d** Apoptosis of activated TCRP1A CD8[+] T cells upon co-incubation for 24 h at the indicated ratio with PMN-MDSC isolated from induced Amela TiRP tumors. Purity of PMN-MDSC cells: 80–90%. Five independent experiments, each in duplicate. **e** Apoptosis of activated TCRP1A CD8[+] T cells was prevented by adding soluble Fas-Fc (10 μg/ml) to PMN-MDSC 1 h before and during the 24 h co-incubation with activated TCRP1A CD8[+] T cells. Three independent experiments, each in duplicate. **f** Ex vivo analysis of apoptosis of TCRP1A CD8[+] T cells in induced Amela TiRP tumors 4 days after adoptive transfer, in mice that were depleted of Ly6G[h] cells by intra-tumoral injection of anti-Ly6G antibody ($n = 42$) or isotype control ($n = 33$) (3 injections of 200 μg every 3 days, starting 4 days before adoptive transfer). The *right panel* shows the efficiency of depletion in the same mice. Results are expressed as mean + s.e.m. Unpaired *t*-test, two-tailed **b**–**e**. *$P < 0.05$, **$P < 0.01$, ***$P < 0.001$. ****$P < 0.0001$

tumors as compared to Mela TiRP tumors, which recruit much less MDSC[3, 4]. Interestingly, overexpression of CCL2 and CCL7 was also reported in human metastatic melanomas that resist anti-PD1 therapy, and is a cardinal feature of the IPRES signature[5].

Having shown that TIL apoptosis in the TiRP model was mediated by FasL, we then analyzed FasL expression on cells from the tumor tissue to determine which cell type was responsible for inducing TIL apoptosis. We consistently observed high levels of FasL on MDSC (Fig. 7a, b). Lower levels of FasL were also observed on endothelial cells and, although inconsistently, on tumor cells. This expression profile was identical whether we analyzed tumors collected before or after ACT (Fig. 7a, b). FasL expression was equally high on PMN-MDSC and M-MDSC (Fig. 7c), but only the former were enriched in induced as compared with transplanted tumors (Fig. 6a). This result suggested that PMN-MDSC were responsible for TIL apoptosis. This conclusion was supported by in vitro experiments showing that PMN-MDSC induced apoptosis of co-cultured TCRP1A CD8[+] T cells, the extent of which was reduced by FasL neutralization with soluble Fas-Fc (Fig. 7d, e). Moreover, TIL apoptosis was reduced in vivo when we depleted PMN-MDSC by intra-tumoral injection of anti-Ly6G antibody (Fig. 7f).

To determine whether in vivo FasL neutralization could increase the efficacy of ACT, we treated mice bearing induced tumors with soluble Fas-Fc starting one week before ACT, and followed tumor growth. We observed better tumor control in mice receiving ACT combined with Fas-Fc, as compared with mice receiving ACT alone (Fig. 8a, b). Even though Ly6G-antibody mediated depletion of PMN-MDSC combined with ACT showed some effect (Fig. 8b), when combined with Fas-Fc and ACT, PMN-MDSC depletion did not further improve tumor control (Fig. 8a, b), indicating that FasL-mediated apoptosis was the dominant suppressive mechanism of PMN-MDSC in this model system. Those results indicate that FasL neutralization can improve the efficacy of immunotherapy based on ACT.

To explore whether FasL neutralization can also increase the efficacy of immunotherapy based on immune checkpoint inhibitors, we first set up experiments in which we treated mice bearing induced TiRP tumors with ACT combined with anti-CTLA4 and anti-PD1 antibodies. The addition of immune checkpoint inhibitors failed to improve tumor rejection, unless it was combined with FasL neutralization and Ly6G-depletion (Fig. 8c). This improved tumor control was, however, similar to the one observed with ACT combined with Fas-Fc ± anti-Ly6G (Fig. 8a, b). These results indicate that TiRP tumors display

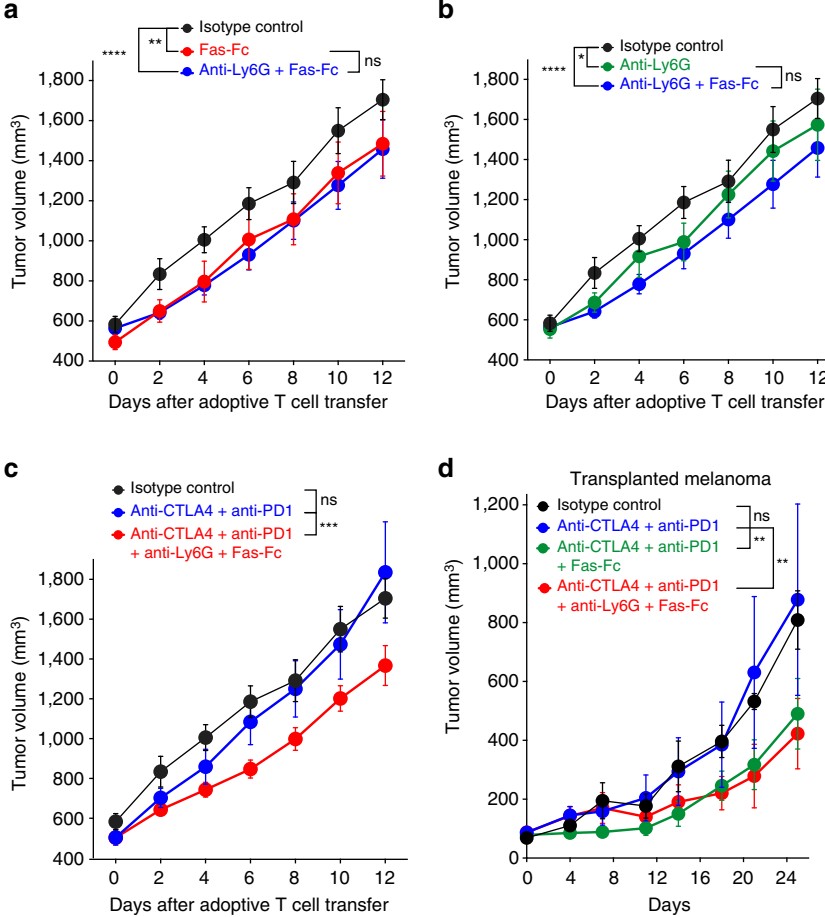

**Fig. 8** FasL neutralization increases the efficacy of immunotherapy. **a**, **b** Mice bearing Amela TiRP tumors received i.p. injections of soluble Fas-Fc and/or anti-Ly6G antibody, starting when tumor size reached 500 mm³, and repeated twice a week. ACT was applied 3 days after the first injection, and tumor volume was monitored (Mean ± s.e.m, n = 10 mice/group). **c** Mice treated as in (**a**, **b**) received anti-CTLA4 and anti-PD1 antibodies i.p. 1 day after the first injection of Fas-Fc/anti-Ly6G, and then twice a week for a total of four injections (Mean ± s.e.m; isotype: n = 10; CTLA4/PD1: n = 9; CTLA4/PD1/Ly6G/Fas-Fc: n = 10; this experiment was run together with the one described in (**a**, **b**) and the isotype control group is identical). **d** Mice bearing transplanted melanomas T429.11 received i.p. injections of anti-Ly6G and/or Fas-Fc starting when tumors became palpable, and repeated twice a week. 1 day later they received i.p. injections of anti-CTLA4 and anti-PD1, repeated twice a week for a total of four injections (Mean ± s.e.m; isotype: n = 7; CTLA4/PD1: n = 7; CTLA4/PD1/Fas-Fc: n = 10, CTLA4/PD1/Ly6G/Fas-Fc: n = 10; data pooled from two independent experiments). Depletion of Ly6G⁺ cells in tumors was checked after killing **a–d**. Two-way ANOVA **a–d**, *P < 0.05, **P < 0.01, ***P < 0.001, ****P < 0.0001

additional immunosuppressive mechanisms that make them insensitive to immune checkpoint inhibitors. To further explore potential synergy between immune checkpoint inhibitors and FasL neutralization, we treated mice bearing transplanted T429.11 tumors with anti-CTLA4 and anti-PD1. We observed no tumor growth inhibition (Fig. 8d). However, when we combined anti-CTLA4 and anti-PD1 with Fas-Fc we observed a significant tumor growth inhibition (P = 0.0004, two-way ANOVA). When we also depleted PMN-MDSC in those mice we observed no further increased tumor growth inhibition, indicating that MDSC do not contribute other major immunosuppressive mechanisms in this setting. Although their recruitment is lower than in induced TiRP tumors, MDSC are also present in transplanted T429.11 tumors (Fig. 6a) and they express high levels of FasL (FasL MFI was 2879 and 2758 for transplanted and induced tumors, respectively, mean of 10 mice each). Therefore, they seem to contribute to resistance to immune checkpoint therapy, although it is possible that other FasL-expressing cells also contribute. Altogether, these results indicate that FasL neutralization has the potential to improve the efficacy of immunotherapy based not only on adoptive cell therapy but also on immune checkpoint inhibitors.

A striking feature of this immunosuppressive mechanism is its antigen-specific nature: only anti-tumor CD8⁺ T cells are affected by apoptosis and disappear. This specificity likely results from the fact that antigenic activation strongly increases Fas expression at the surface of CD8⁺ T cells. This was confirmed when we isolated spleen cells from TiRP-tumor-bearing mice and incubated them with L1210.P1A.B7-1 cells to activate P1A-specific T cells: after 48 h, we compared Fas expression on CD8⁺ T cells that were P1A-specific or not (Supplementary Fig. 10a). Fas expression was much higher on P1A-specific CD8⁺ T cells, which were the only CD8⁺ T cells that were activated in these experimental conditions, as indicated by CD69 expression. Moreover, when we incubated these T cells with FasL, only P1A-specific CD8⁺ T cells underwent apoptosis (Supplementary Fig. 10b). We also observed that IFNγ further increased Fas expression on activated CD8⁺ T cells (Supplementary Fig. 4c). The latter finding likely contributes to explain the reduced TIL apoptosis we observed in mice treated with IFNγ-neutralizing antibodies (Fig. 4c). As FasL is already expressed by MDSC before ACT, and therefore probably does not depend on IFNγ produced by TIL, the pro-apoptotic effect of IFNγ in this setting likely results from increased expression, on activated T cells themselves, of both Fas

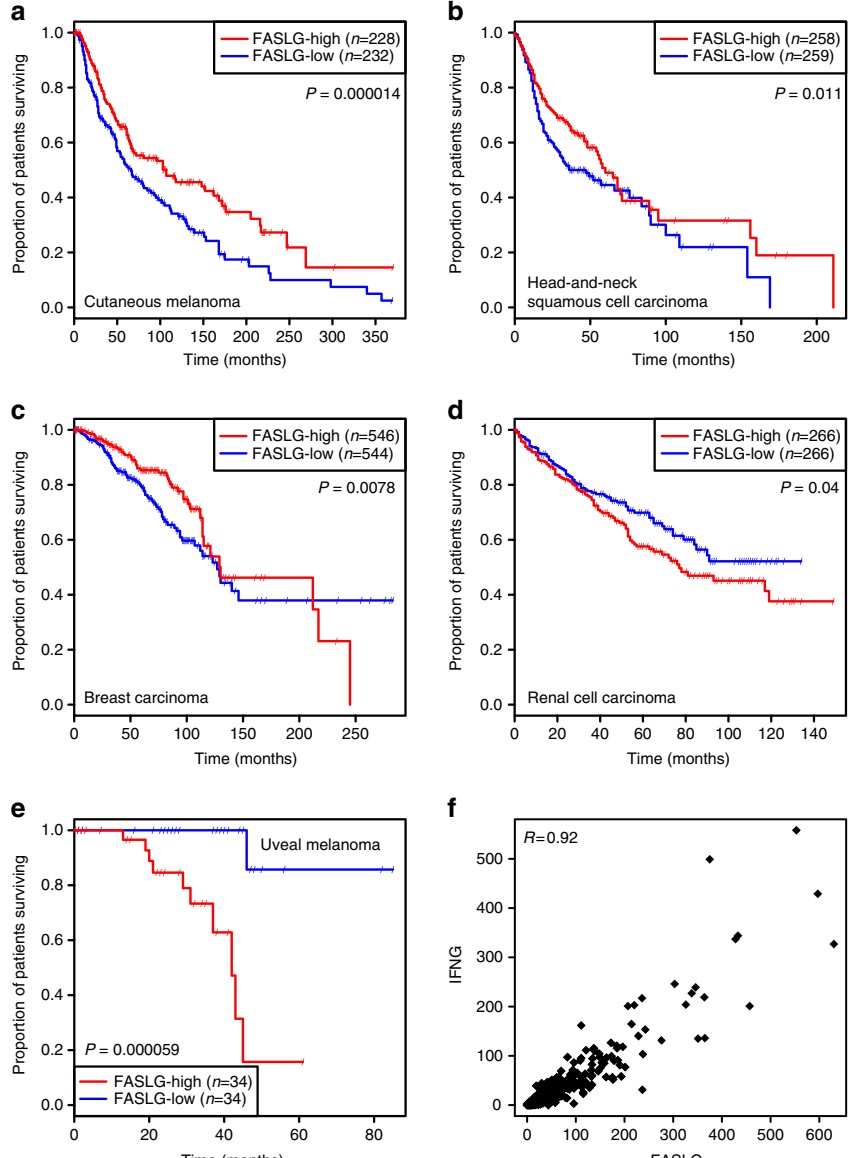

**Fig. 9** Correlation between Fas-ligand expression in human tumors and patient survival. Survival curves of patients with: **a** cutaneous melanoma, **b** head-and-neck squamous cell carcinoma, **c** breast carcinoma, **d** renal cell carcinoma and **e** uveal melanoma, according to high (*red line*) and low (*blue line*) tumoral expression of the *FASLG* gene. Leaning bars indicate censored cases. The survival curves of the two groups were compared using Cox proportional hazard regression. Only the tumor types with significant survival difference are shown. **f** Correlation between *FASLG* and *IFNG* transcript levels in the TCGA melanoma samples (*n* = 469). Each dot represents a tumor sample. *X* and *Y* values indicate RPKM-normalized transcript numbers. Pearson's coefficient of correlation (*R*) is shown

and FasL, according to the well-described phenomenon of activation-induced cell death (AICD), an immune checkpoint process that prevents excessive T-cell activity by inducing Fas/FasL-mediated suicide/fratricide killing of activated T cells[28, 29]. AICD, however, is expected to occur in transplanted tumors, as well as in induced tumors, and therefore cannot account for the increased TIL apoptosis observed in induced tumors, which rather appears to be triggered by FasL-expressing MDSC that are enriched in these tumors.

**Relevance to human tumors**. To determine whether *FASLG* expression was associated with disease progression in human tumors, we used The Cancer Genome Atlas (TCGA) database to compare the survival of patients bearing tumors expressing different levels of FasL. In most tumor types, high *FASLG* transcript levels were associated with a relatively better survival than low levels. This difference was statistically significant in cutaneous melanoma (*P* = 0.000014), head-and-neck squamous cell carcinoma (*P* = 0.011) and breast carcinoma (*P* = 0.0078) (Fig. 9a–c). In sharp contrast, high *FASLG* expression in renal cell carcinoma (*P* = 0.04) and uveal melanoma (*P* = 0.000059) was associated with significantly worse prognosis (Fig. 9d, e). It is noteworthy that these two tumor types are also those that diverge from most other malignancies by their shorter survival associated with higher TIL infiltration[30–32]. We therefore considered the possibility that *FASLG* expression in human tumors was in fact associated with T-cell infiltration. Consistently, expression of *FASLG* in the main TCGA tumor types was strongly correlated with the levels of T-cell-specific transcripts such as *IFNG* (shown for melanoma in Fig. 9f), *CD3E* and *CD8B* (not shown). *FASLG* transcript levels in tumors thus reflect TIL abundance and

activity, in line with the selective expression of FasL in activated T cells, and cannot be used as an independent prognostic factor. Interestingly, a similar correlation with TIL infiltration was observed for the transcript levels of IDO1 and PD-L1, two well-known immune checkpoints that are induced by T-cell activation and involved in adaptive tumoral resistance, as ascertained by the clinical benefit obtained with specific inhibitors[12].

## Discussion

Collectively, our results establish a dominant mechanism of resistance to anti-tumor immunity in the TiRP model of inflammatory melanoma, characterized by the induction of TIL apoptosis through FasL expression by PMN-MDSC infiltrating the tumor. Our results also illustrate, for the first time, the strikingly different behavior of autochthonous vs. isogenic transplanted tumors, which are efficiently controlled by immunotherapy. The long-standing interaction between the host and the autochthonous tumor growing progressively among normal tissues results in a state of abnormal inflammation[4], which allows immunological tolerance of the tumor and is characterized by enrichment in PMN-MDSC. Those cells express high amounts of FasL, which then induces apoptosis of tumor-specific CD8+ T lymphocytes infiltrating the tumor and their progressive depletion from the periphery. These results not only confirm the potential of targeting PMN-MDSC to reduce tumor-induced immunosuppression, but also identify FasL as a promising target whose neutralization could increase the clinical success of cancer immunotherapy.

The role of FasL in mediating tumoral immune resistance has been discussed for many years, following an initial report about FasL expression by melanoma cells triggering TIL apoptosis[33]. Several reports subsequently showed that enforced expression of high-levels of FasL in tumor cells induced inflammation associated with recruitment and activation of neutrophils resulting in tumor rejection[34, 35]. These conflicting results initiated what was called the Fas-counterattack controversy[36]. The issue was further complicated by the poor specificity of the anti-FasL antibodies available at the time, the relative role of membrane-bound vs. soluble FasL, which does not trigger apoptosis, and the interference of FasL expression by other cells, including the T cells themselves in the process of AICD[36–38]. Subsequent work partly clarified the controversy, by suggesting that the pro-inflammatory role of FasL was dependent on high expression levels obtained by enforced FasL expression, and showing that it was not observed with tumors expressing FasL naturally, in which case FasL downregulation reduced tumorigenicity, in line with the immunosuppressive role of FasL[39]. Another key aspect is the type of cells expressing FasL: contrary to the previous work discussed above, in the TiRP model we did not so much observe FasL expression on tumor cells but rather on PMN-MDSC.

Similarly, recent work highlighted the immunosuppressive role of FasL expressed by endothelial cells in ovarian cancer[40]. In this work, FasL-mediated apoptosis of anti-tumor T cells was shown to prevent access of CD8+ T cells to tumor nests in ovarian carcinoma[40]. This was linked to FasL expression by tumor endothelia, inducing apoptosis of T cells reaching the tumor vasculature, thereby preventing T-cell extravasation. Despite similarity in the mode of T-cell execution, the mechanism of our model is different, because TIL extravasation and infiltration is not prevented, but TILs are induced to apoptosis by PMN-MDSC within tumor nests. In practice, however, our observations concur with those of Motz et al. in defining FasL as a relevant drug target whose neutralization could improve the efficacy of cancer immunotherapy. A recent study highlighted an additional benefit of neutralizing FasL in the context of ACT[41]. This study

compared the anti-tumor efficacy of ACT making use of naive or memory anti-tumor CD8+ T cells, or a mix of both. The latter condition would mimic the clinical situation in cancer patients. It is known that naive CD8+ T cells are more efficient due to better persistence after ACT. The authors showed that when naive CD8+ T cells are mixed with memory CD8+ T cells and used for ACT, they undergo a precocious differentiation that limits their anti-tumor efficacy. This precocious differentiation is driven by memory CD8+ T cells through non-apoptotic Fas signaling: FasL neutralization prevents this precocious differentiation of naive CD8+ T cells and increases the efficacy of ACT in this setting. We only used naive CD8+ T cells and injected them after short-term in vitro activation, so this effect does not explain the beneficial effect of FasL neutralization in our experimental setting. But it provides an additional rationale for FasL neutralization in clinical protocols of ACT, in which the autologous CD8+ T cells used are usually a mix of naive and memory CD8+ T cells.

Although the dominant mechanism of T-cell cytotoxicity relies on delivery of apoptosis-inducing granzymes into target cells[42], the FasL pathway also contributes, mostly to bystander cell killing. Such killing of stromal cells through FasL was found to be required for complete tumor rejection in the context of ACT[43]. However, the major role of the FasL pathway in the immune system is to promote activation-induced cell death, which entails fratricide and/or suicide death of T cells induced to express high levels of Fas and FasL upon activation. This counter-regulatory mechanism prevents excessive T-cell activity, and as such qualifies as an immune checkpoint mechanism, whose inactivation should benefit cancer immunotherapy. This notion is supported by the phenotype of gld and lpr mice, which are genetically deficient in FasL and Fas, respectively, and display autoimmune phenotypes[44–47].

In sum, the potential benefits of FasL neutralization for cancer immunotherapy, particularly in the context of adoptive cell therapy, are fourfold: (i) preventing TIL apoptosis induced by FasL-expressing PMN-MDSC, (ii) improving CD8+ T-cell infiltration into the tumor by preventing apoptosis induced by endothelial cells, (iii) improving T-cell persistence at the tumor site by inhibiting the immune checkpoint relying on AICD, (iv) improving T-cell persistence and activity by preventing precocious differentiation of naive T cells induced by co-injected memory T cells. Strategies aimed at neutralizing FasL in combination with clinical approaches of cancer immunotherapy are therefore warranted.

## Methods

**Mice.** TiRP-10B;Ink4a/Arf^flox/flox mice[2] contain an inducible transgene which is controlled by the tyrosinase promoter and drives expression of H-Ras and of Trap1a, which encodes a MAGE-type tumor antigen. They were backcrossed to a B10.D2 background (TiRP mice) and bred to homozygosity (TiRP-10B+/+;Ink4a/Arf^flox/flox). B10.D2;Ink4a/Arf^flox/flox (TiRP-10B−/−) mice were used as controls and as recipients for tumor transplantation experiments. TCRP1A mice heterozygous for the H-2L^d/P1A_{35-43}-specific TCR transgene[10] were kept on the B10.D2;Rag1−/− background. For all the in vivo treatments, mice were randomly divided into groups. Investigators were blinded to the group allocation during the animal experiments. Age-matched littermate mice of both sexes between 5 and 7 weeks of age were used for all in vivo experiments. All mice used in this study were produced under SPF conditions at the animal facility of the Ludwig Institute for Cancer Research. All the rules concerning animal welfare have been respected according to the 2010/63/EU Directive. All procedures were performed with the approval of the local Animal Ethical Committee, with reference 2015/UCL/MD/15.

**Tumor induction with 4OH-Tamoxifen.** A fresh solution of 4OH-Tamoxifen was prepared by dissolving 4OH-Tamoxifen (Sigma Aldrich and Imaginechem) in 100% ethanol and mineral oil (ratio 1:9) followed by 30-min sonication, and injected twice s.c (4 mg/200 µl) in the neck area of anesthetized TiRP mice two weeks apart. The therapeutic experiments were performed exclusively with mice bearing unpigmented Amela tumors. Unless mentioned otherwise, mice with pigmented tumors (Mela) were excluded due to different tumor growth kinetics[4]. Tumor volume (in mm³) was calculated by the following formula: volume=length/

2×width[2]. For occasional mice that developed more than one tumors we only considered the tumor that appeared first.

**Cell lines**. T429.11, T429.6 and T429.18 clones were derived from an induced Amela TiRP tumor referred to as T429[4]. They were cloned from the T429 induced melanoma primary tumor line. The expression of the transgene[2] was evaluated in all three T429 clones and T-cell recognition was tested for each of them in vitro using TCRP1A CD8+ T cells. P511 is an azaguanine-resistant variant of P815[48, 49]. P1204 is a P815AB-negative variant carrying a deletion of gene *Trap1a*[48, 49]. L1210. P1A.B7-1 cells were obtained by transfection of L1210.P1A cells with the murine B7-1 cDNA cloned into plasmid pEFBOS[50]. All cells were maintained at 37 °C with 8% $CO_2$. Unless otherwise specified, all culture media are IMDM (GIBCO) containing 10% fetal bovine serum supplemented with L-arginine (0.55 mM, Merck), L-asparagine (0.24 mM, Merck), glutamine (1.5 mM, Merck), beta-mercaptoethanol (50 μM, Sigma), 50 U ml−1 penicillin and 50 mg ml−1 streptomycin (Life Technologies) (complete medium). Cell lines were tested for mycoplasma contamination.

**Adoptive transfer of TCRP1A CD8+ T cells**. P1A-specific CD8+ T cells were isolated from spleens and lymph nodes of TCRP1A mice using anti-CD8α MicroBeads (Miltenyi Biotec), and stimulated in vitro by co-culture for 4 days with irradiated (10,000 rads) L1210.P1A.B7-1 cells[51] ($10^5$ of each cell type per well in 48-well plates) in IMDM complete medium. After purification on a Ficoll gradient on day 4, $10^7$ living TCRP1A CD8+ T cells were injected i.v. in 200 μl PBS.

**Vaccine and antibodies treatment**. TiRP mice were immunized against P1A-tumor antigen by a heterologous prime-boost regimen consisting of a first injection of an Adeno.Ii.P1At ($10^8$ PFU/mouse/100 μl) followed, 15 days later, by SFV.P1A ($10^7$ IU/mouse/100 μl)[6]. Both viruses were given intradermally. Anti-PD1 (BioXcell, clone RMP1-14, Rat IgG2a, 200 μg/mouse i.p.) and anti-CTLA4 (BioXcell, clone 9H10, Syrian Hamster IgG2, 40 μg/mouse i.p.) treatment were started at the time of tumor appearance every 3 days for a total of 4 injections. Control mice received isotype controls (for CTLA4: BioXcell, Polyclonal Syrian Hamster IgG; for PD-1, BioXcell, Rat IgG2a, clone 2A3).

For the control immunization against P91A (Fig. 1c), mice received 4 intramuscular injections every two weeks of P91A peptide (50 μg) in PBS, mixed v/v with AS15 adjuvant[52] kindly provided by GlaxoSmithKline, Belgium. For in vivo neutralization of IFNγ, mice received 0.5 mg anti-mouse IFNγ (BioXcell, clone XMG1.2,) 1 day before ACT.

For the intra-tumoral depletion of Ly6Gh MDSC, presented in Fig. 7f, mice received 200 μg/mouse of antibody to Ly6G (BioXcell, clone 1A8, Rat IgG2a) or isotype control (BioXcell, Rat IgG2a, clone 2A3) every 3 days starting 4 days before ACT.

For the in vivo antibody treatment combined with adoptive cell transfer (Fig. 8a–c), anti-Ly6G (200 μg/mouse except the first injection which was 400 μg/mouse, twice a week), and Fas-Fc (150 μg/mouse, two injections per week) treatment started when the tumor size reached 500 mm³. The anti-CTLA4 (100 μg/mouse) and anti-PD1 antibodies (200 μg/mouse) were injected 4 times at a 3 days interval i.p. 1 day after the anti-Ly6G and Fc-Fas injection. The corresponding isotype controls were injected accordingly. Mice received $10^7$ activated TCRP1A CD8+ T cells (ACT) 3 days after Fas-Fc and anti-Ly6G injection. Tumor size was monitored every 2 days.

For the in vivo antibody treatments (Fig. 8d), anti-Ly6G (BioXcell, clone 1A8, Rat IgG2a, 200 μg/mouse except the first injection which was 400 μg/mouse, twice a week), and Fas-Fc (150 μg/mouse, two injections per week) treatment started when the tumor size reached 50–150 mm³. The anti-CTLA4 (BioXcell, clone 9H10, 100 μg/mouse) and anti-PD1 antibodies (BioXcell, clone RMP1-14, 200 μg/mouse) were injected 4 times at 3-day intervals i.p. starting 1 day after the anti-Ly6G and Fas-Fc injections. The corresponding isotype controls were injected accordingly. Tumor size was monitored every 2 or 3 days.

**Flow cytometry**. H-2Ld/P1A and P91A tetramers (1 μM) were produced in our Institute[53]. Antibodies used were: CD8α-PerCP (clone 53–6.7, Biolegend, 1 μg/mL); CD69-APC (clone H1.2F3, Biolegend, 1 μg/mL); CD3ε-FITC (clone 17A2, Biolegend, 1 μg/mL); CD45-FITC, CD45-Alexa700 (clone 30-F11, Biolegend, 1 μg/mL); Gr-1-APC, Gr-1-Bv421 (clone RB6-8C5, Biolegend, 1 μg/mL); Ly6C-PerCP (clone HK1.4, Biolegend, 1 μg/mL); Ly6G-PE, Ly6G-Bv510 (clone 1A8, Biolegend, 1 μg/mL); CD11b-BV421 (clone M1/70, Biolegend, 1 μg/mL); CD178 (FasL) (clone Kay-10, Biolegend, 2 μg/mL); CD31-PE/Cy7 (clone 390, Biolegend, 1 μg/mL); CD95-PE/Cy7 (Fas) (clone Jo2, Becton Dickinson, 1 μg/mL) and corresponding isotype controls. Viability dye efluor780 (eBiosciences, 1:500) was used. Annexin V-BV421 or Annexin V-APC (Biolegend, 1:50) was used to monitor apoptosis. P1A antibody (clone P1A102B3, 1 μg/mL) was produced by immunizing P1A-KO mice[54] with a P1A peptide (CEEMGNPDGFSP) coupled to ovalbumin. Hybridomas were sub-cloned and further screened for production of anti-P1A antibodies. Clone P1A102B3 was selected and confirmed to produce an IgG1 recognizing P1A specifically and applicable for western blot, IHC, Flow cytometry and ELISA.

**Quantitative RT-PCR**. Total RNA isolated from tumor tissues were tested by qRT-PCR using the following primers: *Trap1a* (P1A): forward: 5′-AGC-TGA-GGA-AAT-GGG-TGC-TG-3′ (exon 1), reverse: 5′-CAG-CAT-TTT-CAC-ACC-TAC-ACT-CCA-3′ (exon 2), probe: 5′-FAM-CCA-TCA-TTT-AAG-GAA-GAA-TGA-AGT-GAA-GTG-TAG-GAT-GA-TAMRA-3′ (exon 2). *Ifng*: forward: 5′-TCA-AGT-GGC-ATA-GAT-GTG-GAA-GAA-3′, reverse: 5′-TGG-CTC-TGC-AGG-ATT-TTC-ATG-3′, probe: 5′-FAM-TCA-CCA-TCC-TTT-TGC-CAG-TTC-CTC-CAG-TAMRA-3′. Samples were assayed in triplicates and expression levels were normalized to β-actin, which was tested with the following primers: forward actin: 5′-CTC-TGG-CTC-CTA-GCA-CCA-TGA-AG-3′, reverse actin: 5′-GCT-GGA-AGG-TGG-ACA-GTG-AG-3′, probe: 5′-FAM-ATC-GGT-GGC-TCC-ATC-CTG-GC-TAMRA-3′. *Faslg* was amplified using ROX SYBR MasterMix blue dTTP using the following primers: forward: 5′-TGA-ATT-ACC-CAT-GTC-CCC-AG-3′, reverse: 5′-AAA-CTG-ACC-CTG-GAG-CC-3′ (Fig. 4d). In Fig. 4e, *Faslg* was amplified using other primers and probe (Integrated DNA technologies, Mm. PT.58.28447971). *Gapdh* (Mm.PT.39a.1) *Il6* (Mm.PT.58.10005), *Csf1* (Mm. PT.58.11661276), *Csf2* (Mm.PT.58.9186111), *Csf3* (Mm.PT.58.7976206.gs), *Cxcl1* (Mm.PT.58.42076891), *Cxcl12* (Mm.PT.58.12038563), *Ccl2* (Mm.PT.58.42151692), *Tgfb1* (Mm.PT.58.11254750), *Ccl7* (Mm.PT.58.17719534), *Cxcl5* (Mm. PT.58.29518961.g) primer and probe were purchased from Integrated DNA technologies.

**Fas silencing in TCRP1A CD8+ T cells**. Silencing of Fas in activated TCRP1A CD8+ T cells was achieved by using Accell siRNA (1 μM) (Dharmacon), with sequence 5′(P)-GUGCAAGUGCAAACCAGAC(dTdT)-3′. After 1 day of activation, TCRP1A CD8+ T cells were incubated with 1 μM Accell control or Fas siRNA in IMDM complete medium without serum for 24 h, then the medium was replaced with IMDM complete medium with serum. The silencing procedure was repeated on day 3. On day 4, $10^7$ TCRP1A CD8+ T cells were adoptively transferred into mice bearing induced tumors. Silencing efficiency was monitored by Western blot using anti-Fas antibody (Santa Cruz, FL- 335, 1:1000). Blots were stripped and reprobed with a rabbit anti-mouse monoclonal antibody to GAPDH (Cell Signaling, 14C10, 1:5000).

**Production of soluble Fas-Fc**. Recombinant soluble dimers of the murine Fas death receptor extracellular domain (ECD) were produced in Drosophila S2 cells (Invitrogen). The sequence coding for the ECD of murine Fas was fused in frame to the sequence coding for the murine FcIgG2A portion to allow dimerization of the fusion protein. A sequence coding for a derivative of the human IL-9 signal peptide was added at the 5′ end of the fusion gene to allow secretion of the recombinant protein. The recombinant gene was cloned into a derivative of the expression vector pRMHA3, which contains a neomycin-resistance gene and a copper inducible promotor from the Drosophila melanogaster metallothionein gene. Drosophila S2 cells were transfected and stable transfectants selected using 1.5 mg/ml of G418 antibiotic (Roche). Production of the recombinant protein was induced in the cultured cells by the addition of 0.5 mM $CuSO_4$. The protein was purified from the cell culture supernatant by affinity chromatography on a protein A sepharose column (GE Healthcare).

**In vivo apoptosis staining**. Four days after ACT, tumor-bearing mice received an i.v. injection of NIR-FLIVO™ 690 Control (8.5 μg/mouse) or NIR-FLIVO™ 690 (NIR-FLIVO™-DyLight® 690-VAD-FMK in vivo Apoptosis Tracer, ImmunoChemistry Technologies, 8.5 μg/mouse) 4 h before killing. TCRP1A CD8+ T cells were then identified using H-2Ld-P1A tetramers and anti-CD8α antibody and analyzed for FLIVO staining by FACS.

**ELISA**. IFNγ secretion was evaluated by ELISA (Biolegend) in the supernatant of tumor homogenates after 24 h of culture in IMDM medium.

**T-cell apoptosis in organotypic tumor slices in vitro**. Fresh tumor tissue slices of 250–300 μm thickness were prepared using the microtome (LEICA, Vibratome VT1200), and cultured on organotypic inserts for 24 h (Millipore) in 4 ml IMDM complete medium. One million TCRP1A CD8+ T cells pre-labeled with CMAC (CellTracker Blue CMAC Dye) were added for 24 h. After washing and OCT embedding, cryosections (7 μm) of these organotypic tumor tissues were stained for apoptosis using FLICA pan-caspase reagent (ImmunoChemistry Technologies, 1:150) and scanned with MIRAX digital microscope. Quantification was performed using Biopix software.

**MDSC purification**. After mechanical and enzymatic dissociation of induced TiRP tumors, MDSC were enriched by centrifugation on Ficoll-Paque Plus (GE Healthcare). Ly6Gh cells were further purified by magnetic sorting (Miltenyi) and confirmed as PMN-MDSC by FACS analysis (Gr-1h, CD11b+, Ly6Clo and Ly6Gh).

**Tumor transplantation with MDSC isolated from induced tumors**. Single cell suspensions were prepared from induced Amela TiRP tumors and from T429.11 transplanted tumors using gentleMACS™ Dissociator. MDSCs were purified from the induced TiRP tumor suspension (as described above) and mixed with the

transplanted tumor suspension (MDSC/T429.11 ratio: 1/4). The T429.11 tumor cell suspensions mixed or not with isolated MDSC were injected intradermally into naive B10.D2;Ink4a/Arf$^{flox/flox}$ mice, on the left and right flank respectively. When the tumor volume of the left flank reached around 500 mm$^3$, the mice received adoptive cell transfer of TCRP1A specific CD8$^+$ T cells. The tumor size was measured every 4 days from the day of the adoptive cell transfer ($n = 8$ mice).

**MDSC-mediated T-cell suppression assay.** TCRP1A T CD8$^+$ T cells activated in vitro for 4 days with irradiated L1210.P1A.B7-1 cells were purified, washed, counted and 100 μl of 2 × 10$^5$ cells viable cells were seeded in a 96-well plate in IMDM complete medium. Ly6G$^+$ MDSC from induced tumors were isolated and purified as above. TCRP1A CD8$^+$ T cells were co-cultured with Ly6G$^+$ MDSCs at the indicated ratios for 3 days at 37 °C. TCRP1A T cells were then purified using anti-CD8 microbeads. Their proliferation was assessed in an 18-hour $^3$H-Thymidine incorporation assay. TCRP1A CD8$^+$ T cells co-cultured with Ly6G$^+$-MDSC for 3 days as above and purified using anti-CD8 microbeads were tested for cytolytic activity in a standard 4 h-chromium release assay, using P511 cells as P1A$^+$ target cells and P1A$^-$ cells P1204 as cold target competitors.

**Statistics.** Statistical analyses were performed using Prism 6 (GraphPad Software). Comparison between two groups was performed using the unpaired Student $t$-test, while two-way analysis of variance (ANOVA) was used to compare the effects of different treatments on tumor growth. $P$ values < 0.05 were considered significant ($P$-value < 0.05 was annotated *; < 0.01 **; < 0.001 ***; < 0.0001 ****).

The tumoral gene expression values and patient survival data from The Cancer Genome Atlas (TCGA) database were retrieved from the cBioPortal's Cancer Genomic Data Server using the CGDS-R package (http://www.cbioportal.org/cgds_r.jsp) and analyzed with the R software using the "survival" package. The patients were divided in two groups according to high and low FASLG expression level, separated by the median value. Their survival curves were compared using Cox proportional hazard regression model. A $P$-value below 0.05 was considered significant.

**Data availability.** The TCGA data referenced during the study are available in a public repository from the TCGA website (https://cancergenome.nih.gov). The authors declare that all other data supporting the findings of this study are available within the article and its supplementary information files and from the corresponding author upon reasonable request.

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

## Acknowledgements

We thank Guy Warnier, Gilles Gaudray, Julien Gossiaux and Laurent Hermans for large-scale production of TiRP and TCRP1A mice, Nicolas Dauguet for help with flow cytometry, Floriane Ribeiro and Dominique Donckers for help with mouse experiments, Madeleine Swinarska for producing soluble Fas, Pascal Schneider for advice in producing soluble Fas and for providing recombinant FasL, Jacques Van Snick and Pamela Cheou for providing antibodies, Ahmed Essaghir for patient survival analysis, Pierre van der Bruggen and Thierry Boon for critical comments, and Mandy Macharis and Auriane Sibille for editorial assistance. This work was supported by Ludwig Cancer Research, Walloon Excellence in Life Sciences and Biotechnology (WELBIO, Belgium), FNRS-Télévie (Belgium), Foundation Against Cancer (Belgium), de Duve Institute and Université catholique de Louvain (Belgium).

## Author contributions

The data shown were obtained in experiments performed by: J.Z. (Figs.: 2e, 3a, f, g, 4e, 5, 6a, e–g, 7a–e, 8a–c; Supplementary Figs.: 6a, 7b, 8–10), C.P.d.T. (Figs.: 1c, 2c, d, f, 4a, c, d; Supplementary Figs.: 2, 5, 6b, c, 7a), S.C. (Figs.: 1a, b, 2b, 3b–e, h, 4b, 6a–d, 7f; Supplementary Figs. 1, 3, 4), C.L. (Fig. 8d) and N.v.B. (Fig. 9). D.C. prepared the Fas-Fc. A.-M.S.-V. constructed the TiRP line and helped with study design. P.L. prepared the SFV. C.U. supervised some in vivo experiments. B.J.V.d.E. constructed the TiRP line, designed and supervised the study and wrote the paper. All authors were involved in the analysis and interpretation of the data and helped preparing the manuscript.

## Additional information

**Competing interests:** The authors declare no competing financial interests.

