## [Peer Review File · Nature Communications]

Reviewers' comments:

Reviewer #1(Remarks to the Author):

The manuscript describes a novel pathway of resistance to adaptive immunity in the context of cancer immunosurveillance induced by adoptive transfer of P1A-specific T cells (ACT) (and in extension to immune checkpoint blockers) in one GEMM autochthonous melanoma model called "TiRP" (Cre-lox mediated induction of H-Ras and deletion of Ink4A/Arf in melanocytes) and several transplantable clones T426.

The authors demonstrated using several converging model systems that TILs undergo Fas-mediated apoptosis after cognate interaction. Fas is induced following exposure with IFN γ produced by TILs while FasL is expressed by CD11b+Gr1+polymorphonuclear cells (PMN). Depletion of these PMN by anti-Ly6G antibodies ameliorated the ACT-induced antitumor effects. Of note, this regulatory pathway of the adaptive immune response appeared more relevant in unpigmented areas of the melanoma.

Finally, by investigating the TCGA database, the authors show a correlation between FasL transcripts and dismal prognosis in epithelial tumors (N=207) but not in melanoma because of a lack of power of the statistical analysis (N=55).

The overall design of the paper is convincing, and the messages are quite original.

Major comments

In its current form, the paper is not truly convincing at two levels:

1/ The demonstration of the detrimental role of FasL-expressing PMN or Fas molecules in long-term protection against P1A tumor antigens has only been brought up for therapy with adoptive T cell transfer (ACT) but not for treatment with anti-CTLA4 or anti-PD1 Abs (immune checkpoint blockers, ICBs). The authors extrapolate their findings to all immunotherapies, suggesting that IFN γ represents the milestone in this regulatory pathway. However, this assumption is too far-fetched and needs further exemplification using the characterization of individual ICBs or their combination. What are the consequences of infusing Fas-Fc or anti-Ly6G antibodies on the efficacy of anti-CTLA4 or/and anti-PD1 Abs against T426 or TiRP melanomas?

2/ The human data are not very strong. Can the author show that FasL transcripts in human melanoma (or lung) biopsies predict resistance to ipilimumab or nivolumab? This kind of result would significantly improve the level of the paper.

Minor comments

Fig. 5d: Statistics are missing

Fig. 5g: What is the effect of Fas-Fc in this setting?

Reviewer #2(Remarks to the Author):

1) In Figures 2b and 5g the authors show the volume of autochthonous GEMM tumors. How was this measured? Is this a single tumor? What phenotype does this tumor have (e.g. mela –amela)?

2) In Figure 5b mela and amela tumors are shown to differ with respect to PMN-MDSC infiltration. If PMN-MDSC were indeed responsible for inducing apoptosis of tumor-infiltrating T cells, then mela tumors should show less T cell apoptosis than amela tumors?

3) In Figure 5g the authors show that Ab-mediated depletion of PMN-MDSC improves the efficacy of adoptive TC therapy for autochthonous GEMM tumors. However, this does not prove the relevance of increased Fas-mediated apoptosis in tumor-infiltrating T cells for the anti-tumoral treatment efficacy as claimed by the authors. The key experiment would have been to block Fas-FasL interactions, e.g. with Fas-Fc or FasL blocking antibodies and follow the effect on tumor growth. These data would critically strengthen the mechanistic connection proposed by the authors.

4) In ED Figure 8 the authors show a correlation between Fas-ligand expression in tumors and patient survival. What exactly was analysed?

5) The authors briefly discuss the 2014 Nat Med paper by Motz et al. entitled "Tumor endothelium FasL establishes a selective immune barrier promoting tolerance in tumors". However, the role of Fas-FasL interactions for the regulation of anti-tumoral T cell immunity has been controversially discussed for 2 decades. An early paper by Hahne et al entitled "Melanoma cell expression of Fas(Apo-1/CD95) ligand: Implications for tumor immune escape" published in Science 1996 initiated what became to be called the Fas counterattack controversy. The authors should evaluate and show at what levels Fas ligand is expressed by endothelial and melanoma cells under basal and (IFN-driven) inflammatory conditions in their model. Along with a more thorough discussion this would provide a more complete picture of the role of Fas-FasL interactions for tumor-infiltrating T cells.

6) On a more general note: What is the mechanistic connection between the IPRES signature with increased numbers of PMN-MDSCs and FasL-mediated T cell apoptosis?

Reviewer #3(Remarks to the Author):

Powis de Tenbossche and colleagues have studied resistance to immune therapy in a model of P1A-positive melanoma, and demonstrated that one such mechanism relies on loss of tumor-specific TILs through FasL-mediated apoptosis with PMN-MDSC being a major source of FasL. This study addresses a major question in the field of immune therapy, i.e., why does a major fraction of patients with solid tumors not respond to checkpoint inhibitors, and extends recently published data on the so-called innate anti-PD1 resistance signature (Hugo, Cell, 2015). In addition to its clear relevance, study findings are highly novel and the described experiments are well-designed and to-the-point. Manuscript is well written and data are clearly presented.

Suggestions (and concerns I have) are listed below:

1. It is not clear how authors would explain the tumor-specific nature of TIL apoptosis. In case FasL+ PMN-MDSC trigger apoptosis of P1ATCR CD8+ TILs, then Fas expression by CD8+ T cells would expectedly be the decisive factor for apoptosis. Is there reason to assume that P1ATCR CD8+ TILs have higher expression levels of Fas when compared to other CD8+ T cells? Is Fas expression mediated by IFN γ ; but even then why would neighboring T cells not be affected by apoptosis? Authors should look into short experiments to address these two questions; even a negative result is informative. Also, authors should put their finding of killing of tumor-specific TILs into a somewhat broader perspective. For instance, others have found initial tumor-specific tolerance of T cells, which was followed by a more general tolerance of T cells after a long latency period and increase in TGF β levels in serum. Interestingly, Amela tumors are also characterized by a TGF β signature, and it may be worthwhile to measure serum TGF β since lack of such an increase

may explain the lack of a more general tolerance.

2. The difference between transplanted and autochthonous tumors with respect to PMN-MDSC is not completely clear. Why would transplanted tumors contain less PMN-MDSC? Do these tumors express less CCL2, CCL7 or CCL8 mRNA (these data are easy to provide)? Or do these tumors show differences with respect to post-translational modification of these chemokines (Molon, JEM, 2011)? And if so, authors should speculate why this is? In addition, one cannot exclude that PMN-MDSC have multiple actions to limit an effective CD8+ T cell response. Hence, it would be good to know whether the effect of depletion of PMN-MDSC is more limited in case adoptively transferred T cells were silenced for Fas.

3. The relevance of the Fas:FasL pathway is clear (as nicely demonstrated by current manuscript), yet how to interpret such data remains a challenge. For instance, expression of Fas by T cells enhances their sensitivity to apoptosis, whereas expression of FasL by T cells enhances their ability to differentiate towards effector T cells and kill Fas-positive stromal cells (see Klebanoff, JCI, 2016; Listopad, PNAS, 2013). Authors should incorporate and discuss the above papers in their manuscript.

Point by point reply to the reviewers comments

Reviewer #1 (Remarks to the Author):

The manuscript describes a novel pathway of resistance to adaptive immunity in the context of cancer immunosurveillance induced by adoptive transfer of P1A-specific T cells (ACT) (and in extension to immune checkpoint blockers) in one GEMM autochthonous melanoma model called "TiRP" (Cre-lox mediated induction of H-Ras and deletion of Ink4A/Arf in melanocytes) and several transplantable clones T426.

The authors demonstrated using several converging model systems that TILs undergo Fas-mediated apoptosis after cognate interaction. Fas is induced following exposure with IFN γ produced by TILs while FasL is expressed by CD11b+Gr1+polymorphonuclear cells (PMN). Depletion of these PMN by anti-Ly6G antibodies ameliorated the ACT-induced antitumor effects. Of note, this regulatory pathway of the adaptive immune response appeared more relevant in unpigmented areas of the melanoma.

Finally, by investigating the TCGA database, the authors show a correlation between FasL transcripts and dismal prognosis in epithelial tumors (N=207) but not in melanoma because of a lack of power of the statistical analysis (N=55).

The overall design of the paper is convincing, and the messages are quite original.

Major comments

In its current form, the paper is not truly convincing at two levels:

1/ The demonstration of the detrimental role of FasL-expressing PMN or Fas molecules in long-term protection against P1A tumor antigens has only been brought up for therapy with adoptive T cell transfer (ACT) but not for treatment with anti-CTLA4 or anti-PD1 Abs (immune checkpoint blockers, ICBs). The authors extrapolate their findings to all immunotherapies, suggesting that IFN γ represents the milestone in this regulatory pathway. However, this assumption is too far-fetched and needs further exemplification using the characterization of individual ICBs or their combination. What are the consequences of infusing Fas-Fc or anti-Ly6G antibodies on the efficacy of anti-CTLA4 or/and anti-PD1 Abs against T426 or TiRP melanomas?

Answer:

We thank the reviewer for this comment. We have done the proposed experiments. Results obtained with T429 cells confirmed the increased efficacy of anti-CTLA4 and anti-PD1 therapy when combined with Fas-Fc alone or with Fas-Fc combined with anti-Ly6G. This is now shown on new Fig. 7d and discussed on lines 281-302. Interestingly, the addition of anti-Ly6G to Fas-Fc did not increase the benefit observed with Fas-Fc alone, suggesting that in this setting Ly6G+ cells do not have major additional mechanisms of immunosuppression other than FasL. We also tested the combination of anti-CTLA4/anti-PD1 with Fas-Fc/Ly6G in induced TiRP tumors. Although we observed significant tumor control with the full combination as compared to the isotype control, we have chosen not to show those data in the manuscript because we did not feel they were robust enough. This is due to the heterogeneity of growth of autochthonous TiRP tumors, which makes it difficult to obtain homogenous groups in such an experiment. That is one of the reasons we have mostly focused in the manuscript on ACT, which can be applied to homogeneous groups of mice with similar tumor size. When we combined ACT with anti-CTLA4/anti-PD1 therapy (Fig. 7c) we

found increased efficacy by adding Fas-Fc and anti-Ly6G. However, this effect does not go beyond the effect of the combination Fas-Fc/anti-Ly6G with ACT alone. These results indicate that induced TiRP tumors have additional mechanisms of resistance to anti-CTLA4/anti-PD1 therapy. Since we obtained good results in T429 tumors showing the benefit of combining Fas-Fc with anti-CTLA4/PD1 therapy, we felt that this adequately addressed the comment of the reviewer and demonstrated that FasL neutralization can increase the efficacy not only of ACT but also of immune checkpoint inhibitors. Nevertheless, in the discussion and conclusion of our work (lines 411-418) we mostly stress the benefit of FasL neutralization in the context of ACT.

2/ The human data are not very strong. Can the author show that FasL transcripts in human melanoma (or lung) biopsies predict resistance to ipilimumab or nivolumab? This kind of result would significantly improve the level of the paper.

Answer:

We fully agree with the reviewer that such data would strengthen the paper. Unfortunately we are unable to provide such data. We have not generated such results locally, and there are almost no public data available (apart from Hugo et al, Cell 2016, 165:35-44, which concerns only 28 melanoma transcriptomes), presumably because most studies that investigate this association on large series of patients are performed by pharmaceutical companies.

However, as shown on revised Fig. 8, we now show a strong correlation between FasL expression and T cell infiltration in human tumors. This led us to refine our previous findings, and conclude that FasL expression is not an independent factor predicting survival. Rather, like other immune checkpoints such as PDL1 and IDO, its expression correlates with T cell markers, and rather indicates a mechanism of adaptive resistance to immune attack triggered by IFN γ (see also answer to reviewer 2).

Minor comments

Fig. 5d: Statistics are missing

Answer: We have added the statistics to this figure (now Fig. 6c).

Fig. 5g: What is the effect of Fas-Fc in this setting?

Answer:

We agree with the reviewer that it is relevant to test Fas-Fc in the setting of ACT. We have now done this experiment, which is shown on Fig. 7a. The results confirm the benefit of combining ACT with Fas-Fc in induced TiRP tumors. Interestingly, combining Fas-Fc with Ly6G-depletion does not provide additional benefit in this setting either, indicating again that FasL is the dominant immunosuppressive mechanism mediated by MDSC in this model system.

Reviewer #2(Remarks to the Author):

- 1) In Figures 2b and 5g the authors show the volume of autochthonous GEMM tumors. How was this measured? Is this a single tumor? What phenotype does this tumor have (e.g. mela – amela)?

Answer:

These are usually single tumors, with an Amela phenotype. For occasional mice that developed more than one tumors, we only considered the tumor that appeared first. Tumor size was measured with a caliper, and calculated with the formula: volume = length x width²/2. We have clarified these points in the manuscript.

2) In Figure 5b mela and amela tumors are shown to differ with respect to PMN-MDSC infiltration. If PMN-MDSC were indeed responsible for inducing apoptosis of tumor-infiltrating T cells, then mela tumors should show less T cell apoptosis than amela tumors?

Answer:

This is indeed the case, Mela tumors show much less TIL apoptosis, as shown on Fig 3h.

3) In Figure 5g the authors show that Ab-mediated depletion of PMN-MDSC improves the efficacy of adoptive TC therapy for autochthonous GEMM tumors. However, this does not prove the relevance of increased Fas-mediated apoptosis in tumor-infiltrating T cells for the anti-tumoral treatment efficacy as claimed by the authors. The key experiment would have been to block Fas-FasL interactions, e.g. with Fas-Fc or FasL blocking antibodies and follow the effect on tumor growth. These data would critically strengthen the mechanistic connection proposed by the authors.

Answer:

We thank the reviewer for this comment. We agree that it is relevant to test the effect of blocking FasL in combination with adoptive cell therapy for induced TiRP tumors. We have now performed those experiments and replaced data from previous Fig. 5g with new data. We show on new Fig 7a that Fas-Fc does increase the efficacy of ACT to control the growth of induced TiRP tumors. Interestingly, combining Fas-Fc with Ly6G-depletion does not provide additional benefit in this setting, indicating that FasL is the dominant immunosuppressive mechanism mediated by MDSC in this model system.

4) In ED Figure 8 the authors show a correlation between Fas-ligand expression in tumors and patient survival. What exactly was analysed?

Answer:

We thank the reviewer for asking this question: in fact, while reviewing the corresponding data, we noticed that an error, independent of our will, had occurred in the R scripts that were used to generate the survival curves. We apologize for this unwanted mistake and we are grateful to the reviewer for allowing us to correct this data set. The script has now been corrected and the issue has been readdressed.

As shown on revised Fig. 8, we now show a strong correlation between FasL expression and T-cell infiltration in human tumors. This led us to refine our previous findings, and conclude that FasL expression is not an independent factor predicting survival. Rather, like other immune checkpoints such as PDL1 and IDO, its expression correlates with T cell markers, and rather indicates a mechanism of adaptive resistance to immune attack triggered by IFN γ .

The text has been corrected as follows (lines 326-344):

Relevance to human tumors

To determine whether *FASLG* expression was associated with disease progression in human tumors, we used The Cancer Genome Atlas (TCGA) database to compare the survival of patients bearing tumors expressing different levels of FasL. In most tumor types, high *FASLG* levels were associated with a relatively better survival than low *FASLG* transcript levels. This difference was statistically significant in cutaneous melanoma, head-and-neck squamous cell carcinoma and breast carcinoma (Fig. 8a-c). In sharp contrast, high *FASLG* expression in renal cell carcinoma and uveal melanoma was associated with significantly worse prognosis (Fig. 8d, e). It is noteworthy that these two tumor types are also those that diverge from most other malignancies by their shorter survival associated with higher TIL infiltration²⁴⁻²⁶. We therefore considered the possibility that *FASLG* expression in human tumors was in fact associated to T-cell infiltration. Consistently, expression of *FASLG* in the main TCGA tumor types was strongly correlated with the levels of T cell-specific transcripts such as *IFNG* (Fig. 8f), *CD3E* and *CD8B* (not shown). *FASLG* transcript levels in tumors thus reflect TIL abundance and activity, in line with the selective expression of FasL in activated T cells, and cannot be used as an independent prognostic factor. Interestingly, a similar correlation with TIL infiltration was observed for the transcript levels of *IDO1* and *PD-L1*, two well known immune checkpoints that are induced by T-cell activation and involved in adaptive tumoral resistance, as ascertained by the clinical benefit of specific inhibitors¹².

5) The authors briefly discuss the 2014 Nat Med paper by Motz et al. entitled “Tumor endothelium FasL establishes a selective immune barrier promoting tolerance in tumors”. However, the role of Fas-FasL interactions for the regulation of anti-tumoral T cell immunity has been controversially discussed for 2 decades. An early paper by Hahne et al entitled “Melanoma cell expression of Fas(Apo-1/CD95) ligand: Implications for tumor immune escape” published in Science 1996 initiated what became to be called the Fas counterattack controversy. The authors should evaluate and show at what levels Fas ligand is expressed by endothelial and melanoma cells under basal and (IFN-driven) inflammatory conditions in their model. Along with a more thorough discussion this would provide a more complete picture of the role of Fas-FasL interactions for tumor-infiltrating T cells.

Answer:

We have added a comprehensive analysis of FasL expression in tumor stroma, including MDSC, endothelial cells, tumor cells and other cells. The results show significantly higher FasL expression on MDSC as compared to other cells. Endothelial cells and, to some extent, tumor cells also express FasL, but at a lower level. This was not different before and after ACT. See new Fig. 6 ab.

We also expanded the discussion to provide a more complete picture of the role of Fas-FasL interactions for TIL. This discussion covers the FasL counterattack controversy. Please see line 360 onwards.

6) On a more general note: What is the mechanistic connection between the IPRES signature with increased numbers of PMN-MDSCs and FasL-mediated T cell apoptosis?

Answer:

The mechanistic link is the expression of cytokines/chemokines able to recruit MDSC. A cardinal feature of the IPRES signature is the expression of *CCL2* and *CCL7*, two chemokines known for their ability to recruit MDSC. We have now compared the expression of a series of cytokines/chemokines in induced TiRP tumors and in transplanted tumors (new Fig. 5f, g). Most of these factors are overexpressed in induced tumors, including *Ccl2* and *Ccl7*. This confirmed our previous work showing that such cytokines are overexpressed in Amela versus Mela TiRP tumors, which recruit much less MDSC (Soudja et al, 2010 and Wehbe et al, 2012). See text lines 248-258.

We have also reinforced the role of MDSC in this model with a new experiment in which we injected T429 melanoma cells into mice with (left flank) or without (right flank) MDSC isolated from induced TiRP tumors. The ratio was 4 tumor cells for 1 MDSC. We performed Adoptive cell therapy when the left tumor reached 500mm³. The results, shown on new Fig 5e, clearly indicate that MDSC prevent tumor rejection. See lines 242-248.

Reviewer #3(Remarks to the Author):

Powis de Tenbossche and colleagues have studied resistance to immune therapy in a model of P1A-positive melanoma, and demonstrated that one such mechanism relies on loss of tumor-specific TILs through FasL-mediated apoptosis with PMN-MDSC being a major source of FasL. This study addresses a major question in the field of immune therapy, i.e., why does a major fraction of patients with solid tumors not respond to checkpoint inhibitors, and extends recently published data on the so-called innate anti-PD1 resistance signature (Hugo, Cell, 2015). In addition to its clear relevance, study findings are highly novel and the described experiments are well-designed and to-the-point. Manuscript is well written and data are clearly presented.

Suggestions (and concerns I have) are listed below:

1. It is not clear how authors would explain the tumor-specific nature of TIL apoptosis. In case FasL + PMN-MDSC trigger apoptosis of P1ATCR CD8+ TILs, then Fas expression by CD8+ T cells would expectedly be the decisive factor for apoptosis. Is there reason to assume that P1ATCR CD8+ TILs have higher expression levels of Fas when compared to other CD8+ T cells? Is Fas expression mediated by IFN γ ; but even then why would neighboring T cells not be affected by apoptosis? Authors should look into short experiments to address these two questions; even a negative result is informative. Also, authors should put their finding of killing of tumor-specific TILs into a somewhat broader perspective. For instance, others have found initial tumor-specific tolerance of T cells, which was followed by a more general tolerance of T cells after a long latency period and increase in TGF β levels in serum. Interestingly, Amela tumors are also characterized by a TGF β signature, and it may be worthwhile to measure serum TGF β since lack of such an increase may explain the lack of a more general tolerance.

Answer:

We thank the reviewer for these relevant questions, which we have now addressed. We show on Extended Data Fig. 9 that antigenic activation of T cells dramatically increases their surface expression of Fas. This was shown by activating spleen cells from TiRP-tumor bearing mice in vitro with antigenic cells L1210.P1A.B7-1. After 48h we compared Fas expression on P1A-specific (« activated ») and non-P1A-specific CD8 (« non-activated »), which were identified with H2L^d-P1A tetramers and confirmed to be activated by CD69 expression (Extended Data Fig. 9a). We also show that these activated CD8 T cells are more sensitive to apoptosis mediated by FasL (Extended Data Fig. 5b). Lastly we show that IFN γ further increases Fas expression on activated CD8 T cells (Extended Data Fig. 5c). This is now discussed in the text on lines 304-324.

We also further address the possibility of general T cell tolerance, as reported previously in a spontaneous sporadic tumor mouse model (Willimsky et al, 2005; Willimsky et al, 2008). We already showed in Extended Data Fig. 2 that we do not see general T-cell unresponsiveness in the TiRP model, since mice bearing Amela TiRP tumors are perfectly able to mount primary immune responses against unrelated antigens. As suggested by the reviewer, we have now also measured TGF β levels in the serum of mice bearing induced Amela TiRP tumors (new Extended Data Fig. 3).

Although total TGFb levels were increased in the sera of tumor-bearing mice, most of this TGFb was latent, as active TGFb levels were barely detectable. This also supported our conclusion that there is no general T cell tolerance in the TiRP model. See text lines 104-112.

2. The difference between transplanted and autochthonous tumors with respect to PMN-MDSC is not completely clear. Why would transplanted tumors contain less PMN-MDSC? Do these tumors express less CCL2, CCL7 or CCL8 mRNA (these data are easy to provide)? Or do these tumors show differences with respect to post-translational modification of these chemokines (Molon, JEM, 2011)? And if so, authors should speculate why this is? In addition, one cannot exclude that PMN-MDSC have multiple actions to limit an effective CD8+ T cell response. Hence, it would be good to know whether the effect of depletion of PMN-MDSC is more limited in case adoptively transferred T cells were silenced for Fas.

Answer:

As mentioned in the answer to reviewer #2, we have now compared the expression of a series of cytokines/chemokines known to be involved in MDSC recruitment in induced TiRP tumors and transplanted tumors (new Fig. 5f, g). Most of these factors were overexpressed in induced tumors, including Ccl2 and Ccl7. This could explain the recruitment of MDSC in induced Amela TiRP tumors and not in transplanted tumors. It also corroborates our previous observations of increased expression of such factors, including Ccl2 and Ccl7, in Amela TiRP tumors as compared to Mela TiRP tumors, which recruit much less MDSC (Soudja et al, 2010 and Wehbe et al, 2012). See text lines 248-258.

We have also reinforced the role of MDSC in this model with a new experiment in which we injected mice with T429 melanoma cells with (left flank) or without (right flank) MDSC isolated from induced TiRP tumors. The ratio was 4 tumor cells for 1 MDSC. We performed adoptive cell therapy when the left tumor reached 500mm³. The results, shown on new Fig 5e, clearly indicate that MDSC prevent tumor rejection. See lines 242-248.

As pointed by the reviewer, MDSC have multiple actions to limit T cell immunity. This is confirmed by in vitro experiments of co-culture of MDSC with P1ATCR CD8 T cells, which are now shown on Fig. 5c, d (lines 242-244). Yet, when combined with Fas-Fc, PMN-MDSC depletion did not provide any additional benefit in terms of tumor control in TiRP mice receiving ACT. These new results, which are shown on new Fig. 7a and c, indicate that PMN-MDSC do not contribute other major immunosuppressive mechanisms in this experimental setting. See lines 271-302.

3. The relevance of the Fas:FasL pathway is clear (as nicely demonstrated by current manuscript), yet how to interpret such data remains a challenge. For instance, expression of Fas by T cells enhances their sensitivity to apoptosis, whereas expression of FasL by T cells enhances their ability to differentiate towards effector T cells and kill Fas-positive stromal cells (see Klebanoff, JCI, 2016; Listopad, PNAS, 2013). Authors should incorporate and discuss the above papers in their manuscript.

Answer:

We thank the reviewer for this comment. We have now incorporated and discussed these points. Please see revised text on lines 386-409. See also new discussion on the Fas counterattack controversy on lines 360-375.

Reviewers' comments:

Reviewer #1 (Remarks to the Author):

None

Reviewer #3 (Remarks to the Author):

Powis de Tenbossche and colleagues have thoroughly and accurately addressed all concerns.

Reviewer #4 (Remarks to the Author):

The authors adequately addressed the concerns of this reviewer that led to a significant improvement of the manuscript.

However, some points should be clarified.

1. The authors claimed that the immunosuppressive effect against adoptively transferred T cells is mediated by PMN-MDSC, which stronger infiltrate induced tumors than M-MDSC.

However, the authors described increased expression of Ccl2 in these tumors, which is known to recruit predominantly M-MDSC. The authors should discuss the mechanisms of preferential accumulation of PMN-MDSC in tumors.

2. The authors clearly showed the key role of FasL in the immunosuppression induced by PMN-MDSC.

However, the expression of FasL is shown for whole MDSC and not for PMN- and M-MDSC subsets separately (Fig.6a,b). What about the expression of other immunosuppressive molecules (like PD-L1, arginase-1, NO, ROS) in PMN- and M-MDSC subsets?

Please find below the point-by-point reply to the reviewers.

Reviewers' comments:

Reviewer #1 (Remarks to the Author):

None

We thank reviewer #1 for his evaluation and comments about our manuscript.

Reviewer #3 (Remarks to the Author):

Powis de Tenbossche and colleagues have thoroughly and accurately addressed all concerns.

We thank reviewer #3 for his evaluation and comments about our manuscript.

Reviewer #4 (Remarks to the Author):

The authors adequately addressed the concerns of this reviewer that led to a significant improvement of the manuscript.

However, some points should be clarified.

1. The authors claimed that the immunosuppressive effect against adoptively transferred T cells is mediated by PMN-MDSC, which stronger infiltrate induced tumors than M-MDSC.

However, the authors described increased expression of Ccl2 in these tumors, which is known to recruit predominantly M-MDSC. The authors should discuss the mechanisms of preferential accumulation of PMN-MDSC in tumors.

We thank reviewer #4 for raising this point, which was indeed not properly discussed in our manuscript. We have now added a more thorough discussion about the mechanisms of preferential accumulation of PMN-MDSC in induced tumors. This is found on page 12, and reads as follows:

To explain the higher recruitment of PMN-MDSC in induced as compared to transplanted tumors, we measured the expression of a series of cytokines and chemokines known to play a role in the recruitment or differentiation of MDSC^{24,25}. We found that most of them were expressed at a higher level in induced tumors as compared to transplanted tumors (Fig. 6f, g). Among these factors, Csf3, Cxcl1 and Cxcl5 are known to specifically recruit PMN-MDSC²⁴⁻²⁶. Although Ccl2, Cxcl12 and TGF β are mostly known for their ability to recruit M-MDSC, they can also promote accumulation of PMN-MDSC in some tumor settings^{21,24-26}. In addition, it has been suggested that M-MDSC recruited at the tumor site can be subsequently converted into PMN-MDSC in the tumor microenvironment²⁷. These results could therefore explain the accumulation of PMN-MDSC in induced Amela TiRP tumors, and corroborate our previous observations of increased expression of such factors, including Ccl2, Cxcl5 and Ccl17, in Amela TiRP tumors as compared to Mela TiRP tumors, which recruit much less MDSC^{3,4}.

2. The authors clearly showed the key role of FasL in the immunosuppression induced by PMN-MDSC.

However, the expression of FasL is shown for whole MDSC and not for PMN- and M-MDSC subsets separately (Fig.6a,b). What about the expression of other immunosuppressive molecules (like PD-L1, arginase-1, NO, ROS) in PMN- and M-MDSC subsets?

We thank the reviewer for this comment and apologize for omitting to show these data in the first place. We have added the expression of FasL by both PMN-MDSC and M-MDSC subsets. This on Figure 7c, and page 12, and reads as follows:

FasL expression was equally high on PMN-MDSC and M-MDSC (Fig. 7c), but only the former were enriched in induced as compared to transplanted tumors (Fig. 6a).

As requested, we have also now tested the expression of ROS, NO, iNOS, Arginase and PD-L1 on both MDSC subsets, and show the results on new Supplementary Figure 9, with a reference in the text on page 11, which reads as follows:

As compared to M-MDSC, PMN-MDSC produced more ROS, expressed less iNOS and produced less NO, in line with previous reports^{20,21}. They also expressed higher levels of arginase, another immunosuppressive factor that is often expressed in M-MDSC but also in PMN-MDSC²⁰⁻²³ (Supplementary Fig. 9).

We think these additional data appropriately address the reviewer's request and we hope the manuscript will now be considered suitable for publication.

REVIEWERS' COMMENTS:

Reviewer #4 (Remarks to the Author):

The authors performed all necessary experiments to address the questions. There are no concerns anymore.